# Long-Sequence Recommendation Models Need Decoupled Embeddings

**Ningya Feng**[1]*, **Junwei Pan**[2]*, **Jialong Wu**[1]*, **Baixu Chen**[1], **Ximei Wang**[2], **Qian Li**[2], **Xian Hu**[2]
**Jie Jiang**[2], **Mingsheng Long**[1]✉
[1]School of Software, BNRist, Tsinghua University, China [2]Tencent Inc, China
`fny21@mails.tsinghua.edu.cn,jonaspan@tencent.com,wujialong0229@gmail.com`
`mingsheng@tsinghua.edu.cn`

## Abstract

Lifelong user behavior sequences are crucial for capturing user interests and predicting user responses in modern recommendation systems. A two-stage paradigm is typically adopted to handle these long sequences: a subset of relevant behaviors is first searched from the original long sequences via an attention mechanism in the first stage and then aggregated with the target item to construct a discriminative representation for prediction in the second stage. In this work, we identify and characterize, for the first time, a neglected deficiency in existing long-sequence recommendation models: a single set of embeddings struggles with learning both attention and representation, leading to interference between these two processes. Initial attempts to address this issue with some common methods (e.g., linear projections—a technique borrowed from language processing) proved ineffective, shedding light on the unique challenges of recommendation models. To overcome this, we propose the Decoupled Attention and Representation Embeddings (DARE) model, where two distinct embedding tables are initialized and learned separately to fully decouple attention and representation. Extensive experiments and analysis demonstrate that DARE provides more accurate searches of correlated behaviors and outperforms baselines with AUC gains up to 9‰ on public datasets and notable improvements on Tencent's advertising platform. Furthermore, decoupling embedding spaces allows us to reduce the attention embedding dimension and accelerate the search procedure by 50% without significant performance impact, enabling more efficient, high-performance online serving. Code in PyTorch for experiments, including model analysis, is available at `https://github.com/thuml/DARE`.

## 1 Introduction

In recommendation systems, content providers must deliver well-suited items to diverse users. To enhance user engagement, the provided items should align with user interests, as evidenced by their clicking behaviors. Thus, the Click-Through Rate (CTR) prediction for target items has become a fundamental task. Accurate predictions rely heavily on effectively capturing user interests as reflected in their history behaviors. Previous research has shown that longer user histories facilitate more accurate predictions (Pi et al., 2020). Consequently, long-sequence recommendation models have attracted significant research interest in recent years (Chen et al., 2021; Cao et al., 2022).

In online services, system response delays can severely disrupt the user experience, making efficient handling of long sequences within a limited time crucial. A general paradigm employs a two-stage process (Pi et al., 2020): *search* (a.k.a. General Search Unit) and *sequence modeling* (a.k.a. Exact Search Unit). This method relies on two core modules: the *attention* module[1], which measures the target-behavior correlation, and the *representation* module, which generates discriminative representations of behaviors. The search stage uses the attention module to retrieve top-k relevant behaviors,

---

*Equal contribution. Work was done while Ningya Feng and Baixu Chen were interns at Tencent.
[1]In this paper, "attention" refers to attention scores—the softmax output that weights each behavior.

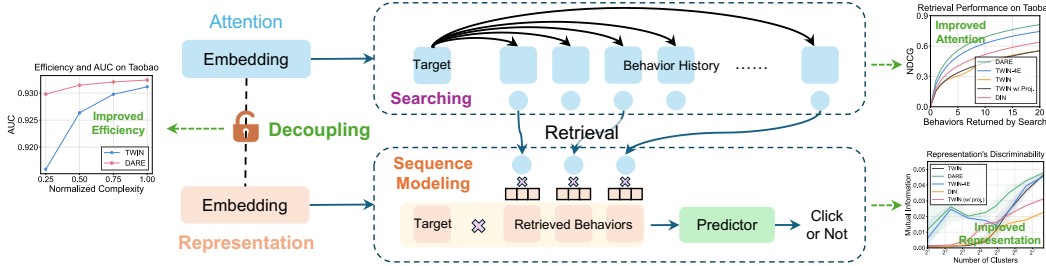

Figure 1: Overview of our work. During search, only a limited number of important behaviors are retrieved according to their attention scores. During sequence modeling, the selected behaviors are aggregated into a discriminative representation for prediction. Our DARE model decouples the embeddings used in attention calculation and representation aggregation, effectively resolving their conflict and leading to improved performance and faster inference speed.

constructing a shorter sub-sequence from the original long behavior sequence[2]. The sequence modeling stage relies on both modules to predict user responses by aggregating behavior representations in the sub-sequence based on their attention, thus extracting a discriminative representation. Existing works widely adopt this paradigm (Pi et al., 2020; Chang et al., 2023; Si et al., 2024).

Attention is critical in the long-sequence recommendation, as it not only models the importance of each behavior for sequence modeling but, more importantly, *determines which behaviors are selected in the search stage*. However, in most existing works, the attention and representation modules share the same embeddings despite serving distinct functions—one learning correlation scores, the other learning discriminative representations. *We analyze these two modules, for the first time, in the perspective of Multi-Task Learning (MTL)*. (Caruana, 1997). Adopting gradient analysis commonly used in MTL (Yu et al., 2020; Liu et al., 2021), we reveal that, unfortunately, gradients of these shared embeddings are dominated by representation, and more concerning, gradient directions from two modules tend to conflict with each other. *Domination and conflict of gradients are two typical phenomena of interference between tasks, influencing the model's performance on both tasks*. Our experimental results are consistent with the theoretical insight: attention fails to capture behavior importance accurately, causing key behaviors to be mistakenly filtered out during the search stage (as shown in Sec. 4.3). Furthermore, gradient conflicts also degrade the discriminability of the representations (as shown in Sec. 4.4).

Inspired by the use of separate query, key (for attention), and value (for representation) projection matrices in the original self-attention mechanism (Vaswani et al., 2017), we experimented with attention- and representation-specific projections in recommendation models, aiming to resolve conflicts between these two modules. However, this approach did not yield positive results. We also tried three other kinds of candidate methods, but unfortunately, none of them worked effectively. Through insightful empirical analysis, we hypothesize that the failure is due to the significantly lower capacity (i.e., fewer parameters) of the projection matrices in recommendation models compared to those in natural language processing (NLP). This limitation is difficult to overcome, as it stems from the low embedding dimension imposed by interaction collapse theory (Guo et al., 2024).

To address these issues, we propose the Decoupled Attention and Representation Embeddings (DARE) model, which completely decouples these two modules at the embedding level by using two independent embedding tables—one for attention and the other for representation. This decoupling allows us to fully optimize attention to capture correlation and representation to enhance discriminability. Furthermore, by separating the embeddings, we can accelerate the search stage by 50% by reducing the attention embedding dimension to half, with minimal impact on performance. On the public Taobao and Tmall long-sequence datasets, DARE outperforms the state-of-the-art TWIN model across all embedding dimensions, achieving AUC improvements of up to 9‰. Online evaluation on Tencent's advertising platform, one of the world's largest platforms, achieves a 1.47% lift in GMV (Gross Merchandise Value). Our contribution can be summarized as follows:

---

[2]The search stage can also be "hard" selecting behaviors by category, but we focus on soft search based on learned correlations for better user interest modeling.

- We identify the issue of interference between attention and representation learning in existing long-sequence recommendation models and demonstrate that common methods (e.g., linear projections borrowed from NLP) fail to decouple these two modules effectively.

- We propose the DARE model, which uses module-specific embeddings to fully decouple attention and representation. Our comprehensive analysis shows that our model significantly improves attention accuracy and representation discriminability.

- Our model achieves state-of-the-art on two public datasets and gets a 1.47% GMV lift in one of the world's largest recommendation systems. Additionally, our method can largely accelerate the search stage by reducing decoupled attention embedding size.

## 2  AN IN-DEPTH ANALYSIS INTO ATTENTION AND REPRESENTATION

In this section, we first review the general formulation for long-sequence recommendation. Then, we analyze the training of shared embeddings, highlighting the domination and conflict of gradients from the attention and representation modules. Finally, we explore why straightforward approaches (e.g., using module-specific projection matrices) fail to address the issue.

### 2.1  PRELIMINARIES

**Problem formulation.**  We consider the fundamental task, Click-Through Rate (CTR) prediction, which aims to predict whether a user will click a specific target item based on the user's behavior history. This is typically formulated as binary classification, learning a predictor $f : \mathcal{X} \mapsto [0,1]$ given a training dataset $\mathcal{D} = \{(\mathbf{x}_1, y_1), \ldots, (\mathbf{x}_{|\mathcal{D}|}, y_{|\mathcal{D}|})\}$, where $\mathbf{x}$ contains a sequence of items representing behavior history and another single item representing the target.

**Long-sequence recommendation model.**  To satisfy the strictly limited inference time in online services, current long-sequence recommendation models generally construct a short sequence first by retrieving top-k correlated behaviors. The attention scores are measured by the scaled dot product of behavior and target embedding. Formally, the $i$-th history behavior and target $t$ is embedded into $\boldsymbol{e}_i$ and $\boldsymbol{v}_t \in \mathbb{R}^{\tilde{d}}$, and without loss of generality, $1, 2, \ldots, K = \text{Top-K}(\langle \boldsymbol{e}_i, \boldsymbol{v}_t \rangle, i \in [1, N])$, where $\langle \cdot, \cdot \rangle$ stands for dot product. Then the weight of each behavior $w_i$ is calculated using softmax function: $w_i = \frac{e^{\langle \boldsymbol{e}_i, \boldsymbol{v}_t \rangle / \sqrt{d}}}{\sum_{j=1}^{K} e^{\langle \boldsymbol{e}_j, \boldsymbol{v}_t \rangle / \sqrt{d}}}$. Finally, the representations of retrieved behaviors are compressed into $\boldsymbol{h} = \sum_{i=1}^{K} w_i \cdot \boldsymbol{e}_i$. TWIN (Chang et al., 2023) follows this structure and achieves state-of-the-art performance through exquisite industrial optimization.

### 2.2  GRADIENT ANALYSIS OF DOMINATION AND CONFLICT

The attention and representation modules can be seen as two tasks: the former focuses on learning correlation scores for behaviors, while the latter focuses on learning discriminative (i.e., separable) representations in a high-dimensional space. However, current methods use a shared embedding for both tasks, which may cause a similar phenomenon to "task conflict" in Multi-Task Learning (MTL) (Yu et al., 2020; Liu et al., 2021) and prevent either from being fully achieved. To validate this assumption, we analyze the gradients from both modules on the shared embeddings.

**Experimental validation.**  Following the methods in MTL, we empirically observe the gradients back propagated to the embeddings from the attention and representation modules. Comparing their gradient norms, we find that gradients from the representation are *five times* larger, **dominating** those from attention, as demonstrated in Fig. 2. Observing their gradient directions, we further find that in nearly two-thirds of cases, the cosine of the gradient angles is negative, indicating the **conflict** between them, as shown in Fig. 3. Domination and conflict are two typical phenomena of task interference, suggesting challenges in learning them well.

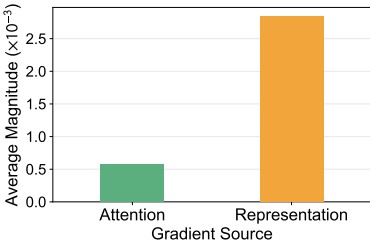

Figure 2: The magnitude of embedding gradients from the attention and representation modules.

In summary, the attention module and representation modules optimize the embedding table towards different directions with varying intensities during training, causing attention to lose correlation accuracy and representation to lose its discriminability. Notably, due to domination, such influence is more severe to attention, as indicated by the poor learned correlation between categories in Sec. 4.3. While some commonly used techniques in MTL may ease the conflict, we tend to seek an optimized model structure that further resolves the conflict.

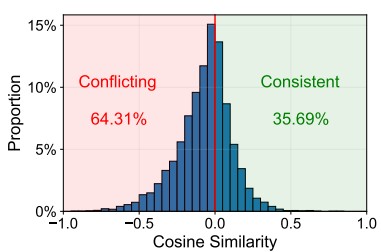

Figure 3: Cosine angles of gradients.

*Finding 1. In sequential recommenders, gradients from the representation module tend to conflict with that from the attention module, and typically dominate the embedding gradients.*

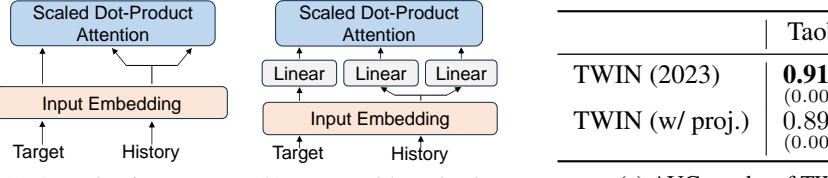

| | Taobao | Tmall |
|---|---|---|
| TWIN (2023) | **0.91688** (0.00211) | 0.95812 (0.00073) |
| TWIN (w/ proj.) | 0.89642 (0.00351) | **0.96152** (0.00088) |

(a) Attention in TWIN    (b) TWIN with projection    (c) AUC results of TWIN variants

Figure 4: Illustration and evaluation for adopting linear projections. (a-b) The attention module in the original TWIN and after adopting linear projections. (c) Performance of TWIN variants. Adopting linear projections causes an AUC drop of nearly 2% on Taobao.

## 2.3 RECOMMENDATION MODELS CALL FOR MORE POWERFUL DECOUPLING METHODS

**Normal decoupling methods fail to resolve conflicts.** To address such conflict, a straightforward approach is to use separate projections for attention and representation, mapping the original embeddings into two new decoupled spaces. This is adopted in the standard self-attention mechanism (Vaswani et al., 2017), which introduces query, key (for attention), and value projection matrices (for representation). Inspired by this, we propose a variant of TWIN that utilizes linear projections to decouple attention and representation modules, named TWIN (w/ proj.). The comparison with the original TWIN structure is shown in Fig. 4a and 4b. Surprisingly, *linear projection, which works well in NLP, loses efficacy in recommendation systems*, leading to negative performance impact, as shown in Tab. 4c. We also tried three kinds of other candidate methods (MLP-based projection, strengthening the capacity of linear projection, and gradient normalization), resulting in a total of eight models, but none of them resolved the conflict effectively. For the structure of these models and more details, refer to Appendix C.

**Larger embedding dimension makes linear projection effective in NLP.** The failure of introducing projection matrices makes us wonder why it works well in NLP but not in recommendation. One possible reason is that the relative capacity of projection matrices regarding the token numbers in NLP is usually strong, *e.g.*, with an embedding dimension of 4096 in LLaMA3.1 (Dubey et al., 2024), there are around 16 million parameters $(4096 \times 4096 = 16,777,216)$ in each projection matrix to map only 128,000 tokens in the vocabulary. To validate our hypothesis, we conduct a synthetic experiment in NLP using nanoGPT (Andrej) with the Shakespeare dataset. In particular, we decrease its embedding dimension from 128 to 2 and check the performance gap be-

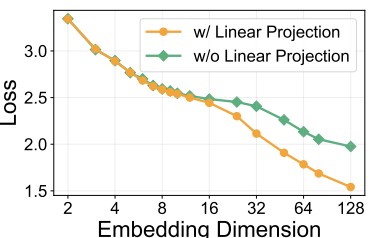

Figure 5: The influence of linear projections with different embedding dimensions in NLP.

tween the two models with/without projection matrices. As shown in Fig. 5, we observe that when the matrix has enough capacity, *i.e.*, the embedding dimension is larger than 16, projection leads to significantly less loss. However, when the matrix capacity is further reduced, the gap vanishes. Our experiment indicates that using projection matrices only works with enough capacity.

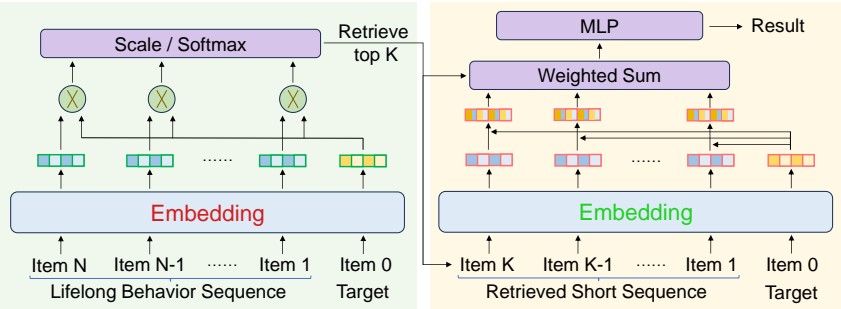

Figure 6: Architecture of the proposed DARE model. One embedding is responsible for attention, learning the correlation between the target and history behaviors, while another embedding is responsible for representation, learning discriminative representations for prediction. Decoupling these two embeddings allows us to resolve the conflict between the two modules.

**Limited embedding dimension makes linear projections fail in recommendation.** In contrast, due to the interaction collapse theory (Guo et al., 2024), the embedding dimension in recommendation is usually no larger than 200, leading to only up to 40000 parameters for each matrix to map millions to billions of IDs. Therefore, *the projection matrices in recommendation never get enough capacity, making them unable to decouple attention and representation.* In this case, other normal decoupling methods mentioned in Appendix C also suffer from weak capacity.

> *Finding 2. Normal methods like linear projection fail to decouple attention and representation in sequential recommendation models due to limited capacity caused by embedding dimension.*

## 3 DARE: DECOUPLED ATTENTION AND REPRESENTATION EMBEDDINGS

With all eight normal decoupling models shown in Appendix C failed, based on our analysis, we seek methods with enough capacity, hoping to completely resolve the conflict. To this end, we propose to decouple these two modules at the embedding level. That is, we employ two embedding tables, one for attention ($E^{\text{Att}}$) and another for representation ($E^{\text{Repr}}$). With gradient back propagated to different embedding tables, our method has the potential to fully resolve the gradient domination and conflict between these two modules. We introduce our model specifically in this section and demonstrate its advantage by experiments in the next section.

### 3.1 ATTENTION EMBEDDING

Attention measures the correlation between history behaviors and the target (Zhou et al., 2018). Following the common practice, we use the scaled dot-product function (Vaswani et al., 2017). Mathematically, the $i$-th history behavior $i$ and target $t$, are embedded into $e_i^{\text{Att}}, v_t^{\text{Att}} \sim E^{\text{Att}}$, where $E^{\text{Att}}$ is the attention embedding table. After retrieval $1, 2, \ldots, K = \text{Top-K}(\langle e_i, v_t \rangle, i \in [1, N])$ their weight $w_i$ is formalized as:

$$w_i = \frac{e^{\langle e_i^{\text{Att}}, v_t^{\text{Att}} \rangle / \sqrt{|E^{\text{Att}}|}}}{\sum_{j=1}^{K} e^{\langle e_j^{\text{Att}}, v_t^{\text{Att}} \rangle / \sqrt{|E^{\text{Att}}|}}}, \tag{1}$$

where $\langle \cdot, \cdot \rangle$ stands for dot product and $|E^{\text{Att}}|$ stands for the embedding dimension.

### 3.2 REPRESENTATION EMBEDDING

In the representation part, another embedding table $E^{\text{Repr}}$ is used, where $i$ and $t$ is embedded into $e_i^{\text{Repr}}, v_t^{\text{Repr}} \sim E^{\text{Repr}}$. Most existing methods multiply the attention weight with the representation of each retrieved behavior and then concatenate it with the embedding of the target as the input of Multi-Layer Perceptron (MLP): $[\sum_i w_i e_i, v_t]$. However, it has been proved that MLP struggles to effectively learn explicit interactions (Rendle et al., 2020; Zhai et al., 2023). To enhance the discrim-

inability, following TIN (Zhou et al., 2024), we adopt the target-aware representation $e_i^{\text{Repr}} \odot v_t^{\text{Repr}}$, denoted as TR in our following paper (refer to Sec. 4.4 for empirical evaluation of discriminability).

The overall structure of our model is shown in Fig. 6. Formally, user history $h$ is compressed into: $h = \sum_{i=1}^{K} w_i \cdot (e_i^{\text{Repr}} \odot v_t^{\text{Repr}})$.

### 3.3 INFERENCE ACCELERATION

By decoupling the attention and representation embedding tables, the dimension of attention embeddings $E^{\text{Att}}$ and the dimension of representation embeddings $E^{\text{Repr}}$ have more flexibility. In particular, we can reduce $E^{\text{Att}}$ while keeping $E^{\text{Repr}}$ to accelerate the searching over the original long sequence whilst not affecting the model's performance. Empirical experiments in Sec. 4.5 show that our model has the potential to speed up searching by 50% with quite little influence on performance and even by 75% with an acceptable performance loss.

### 3.4 DISCUSSION

Considering the superiority of decoupling the attention and representation embeddings, one may naturally raise an idea: we can further decouple the embeddings of history and target within the attention (and representation) module, *i.e.* forming a TWIN with 4 Embeddings method, or TWIN-4E in short, consisting of attention-history (named keys in NLP) $e_i^{\text{Att}} \in E^{\text{Att-h}}$, attention-target (named querys in NLP)

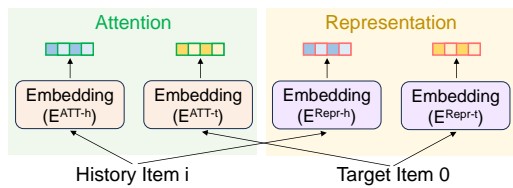

Figure 7: Illustration of the TWIN-4E model.

$v_t^{\text{Att}} \in E^{\text{Att-t}}$, representation-history (named values in NLP) $e_i^{\text{Repr}} \in E^{\text{Repr-h}}$ and representation-target $v_t^{\text{Repr}} \in E^{\text{Repr-t}}$. The structure of TWIN-4E is shown in Fig. 7. Compared to our DARE model, TWIN-4E further decouples the behaviors and the target, meaning that the same category or item has two totally independent embeddings as behavior and target. This is strongly against two prior knowledge in recommendation system. 1. The correlation of two behaviors is similar no matter which is the target and which is from history. 2. Behaviors with the same category should be more correlated, which is natural in DARE since a vector's dot product with itself tends to be bigger.

## 4 EXPERIMENTS

### 4.1 SETUP

**Datasets and task.** We use the publicly available Taobao (Zhu et al., 2018; 2019; Zhuo et al., 2020) and Tmall (Tianchi, 2018) datasets, which provide users' behavior data over specific time periods on their platforms. Each dataset includes the items users clicked, represented by item IDs and their corresponding category IDs. Thus, a user's history is modeled as a sequence of item and category IDs. The model's input consists of a recent, continuous sub-sequence of the user's lifelong history, along with a target item. For positive samples, the target items are the actual items users clicked next, and the model is expected to output "Yes." For negative samples, the target items are randomly sampled, and the model should output "No." In addition to these public datasets, we validated our performance on one of the world's largest online advertising platforms. More details on datasets and training/validation/test splits are shown in Appendix B.

**Baselines.** We compare against a variety of recommendation models, including ETA (Chen et al., 2021), SDIM (Cao et al., 2022), DIN (Zhou et al., 2018), TWIN (Chang et al., 2023) and its variants, as well as TWIN-V2 (Si et al., 2024). As discussed in Sec.3.2, the target-aware representation by crossing $e_i^{\text{Repr}} \odot v_t^{\text{Repr}}$ significantly improves representation discriminability, so we include it in our baselines for fairness. **TWIN-4E** refers to the model introduced in Sec. 3.4, while **TWIN (w/ proj.)** refers to the model described in Sec. 2.3. **TWIN (hard)** represents a variant using "hard search" in the search stage, meaning it only retrieves behaviors from the same category as the target. **TWIN (w/o TR)** refers to the original TWIN model without target-aware representation, *i.e.*, representing user history as $h = \sum_i w_i \cdot e_i$ instead of $h = \sum_i w_i (e_i \odot v_t)$.

Table 1: Overall comparison reported by the means and standard deviations of AUC. The best results are highlighted in bold, while the previous best model is underlined. Our model outperforms all existing methods with obvious advantages, especially with small embedding dimensions.

| Setup | Embedding Dim. = 16 | | Embedding Dim. = 64 | | Embedding Dim. = 128 | |
|---|---|---|---|---|---|---|
| Dataset | Taobao | Tmall | Taobao | Tmall | Taobao | Tmall |
| ETA (2021) | 0.91326 (0.00338) | 0.95744 (0.00108) | 0.92300 (0.00079) | 0.96658 (0.00042) | 0.92480 (0.00032) | 0.96956 (0.00039) |
| SDIM (2022) | 0.90430 (0.00103) | 0.93516 (0.00069) | 0.90854 (0.00085) | 0.94110 (0.00093) | 0.91108 (0.00119) | 0.94298 (0.00081) |
| DIN (2018) | 0.90442 (0.00060) | 0.95894 (0.00037) | 0.90912 (0.00092) | 0.96194 (0.00033) | 0.91078 (0.00054) | 0.96428 (0.00013) |
| TWIN (2023) | 0.91688 (0.00211) | 0.95812 (0.00073) | 0.92636 (0.00052) | 0.96684 (0.00039) | 0.93116 (0.00056) | 0.97060 (0.00005) |
| TWIN (hard) | 0.91002 (0.00053) | 0.96026 (0.00024) | 0.91984 (0.00048) | 0.96448 (0.00042) | 0.91446 (0.00055) | 0.96712 (0.00019) |
| TWIN (w/ proj.) | 0.89642 (0.00351) | 0.96152 (0.00088) | 0.87176 (0.00437) | 0.95570 (0.00403) | 0.87990 (0.02022) | 0.95724 (0.00194) |
| TWIN (w/o TR) | 0.90732 (0.00063) | 0.96170 (0.00057) | 0.91590 (0.00083) | 0.96320 (0.00032) | 0.92060 (0.00084) | 0.96366 (0.00103) |
| TWIN-V2 (2024) | 0.89434 (0.00077) | 0.94714 (0.00110) | 0.90170 (0.00063) | 0.95378 (0.00037) | 0.90586 (0.00059) | 0.95732 (0.00045) |
| TWIN-4E | 0.90414 (0.01329) | 0.96124 (0.00026) | 0.90356 (0.01505) | 0.96372 (0.0004) | 0.90946 (0.01508) | 0.96016 (0.01048) |
| DARE (Ours) | **0.92568** (0.00025) | **0.96800** (0.00024) | **0.92992** (0.00046) | **0.97074** (0.00012) | **0.93242** (0.00045) | **0.97254** (0.00016) |

## 4.2 OVERALL PERFORMANCE

In recommendation systems, it is well-recognized that even increasing AUC by 1‰ to 2‰ is more than enough to bring online profit. As shown in Tab. 1, our model achieves AUC improvements of 1‰ and 9‰ compared to current state-of-the-art methods across all settings with various embedding sizes. In particular, significant AUC lifts of 9‰ and 6‰ are witnessed with an embedding dimension of 16 on Taobao and Tmall datasets, respectively.

There are also some notable findings. TWIN outperforms TWIN (w/o TR) in most cases, proving that target-aware representation $e_i^{\text{Repr}} \odot v_t^{\text{Repr}}$ do help enhance discriminability (further evidence is shown in Sec. 4.4). Our DARE model has an obvious advantage over TWIN-4E, confirming that the prior knowledge discussed in Sec. 3.4 is well-suited for the recommendation system. ETA and SDIM, which are based on TWIN and focus on accelerating the search stage at the expense of performance, understandably show lower AUC scores. TWIN-V2, a domain-specific method optimized for video recommendations, is less effective in our settings.

## 4.3 ATTENTION ACCURACY

Mutual information, which captures the shared information between two variables, is a powerful tool for understanding relationships in data. We calculate the mutual information between behaviors and the target as the ground truth correlation, following (Zhou et al., 2024). The learned attention score reflects the model's measurement of the importance of each behavior. Therefore, we compare the attention distribution with mutual information in Fig. 8.

In particular, Fig. 8a presents the mutual information between a target category and behaviors with top-10 categories and their target-relative positions (i.e., how close to the target is the behavior across time). We observe *a strong semantic-temporal correlation*: behaviors from the same category as the target (5th row) are generally more correlated, with a noticeable temporal decay pattern. Fig. 8b presents TWIN's learned attention scores, which show a decent temporal decay pattern but *overestimate the semantic correlation of behaviors across different categories*, making it too sensitive to recent behaviors, even those from unrelated categories. In contrast, *our proposed DARE can effectively capture both the temporal decaying and semantic patterns*.

The retrieval in the search stage relies entirely on attention scores. Thus, we further investigate the retrieval on the test dataset, which provides a more intuitive reflection of attention quality. Behaviors with top-k mutual information are considered the optimal retrieval, and we evaluate model performance using normalized discounted cumulative gain (NDCG) (Järvelin & Kekäläinen, 2002). The results, along with case studies, are presented in Fig. 9 (more examples in Appendix E.4). We find:

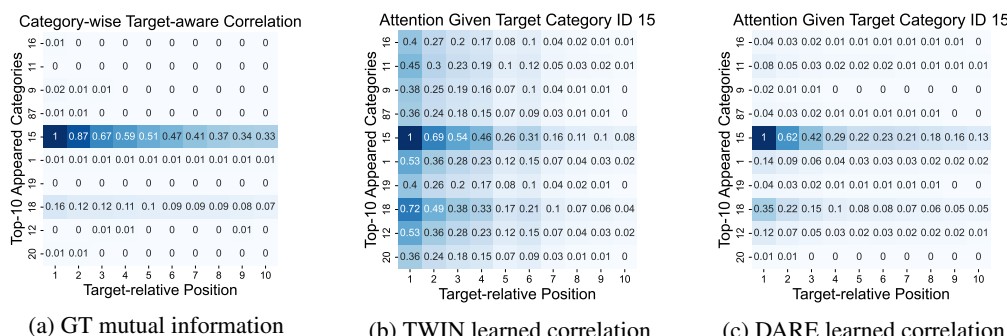

(a) GT mutual information   (b) TWIN learned correlation   (c) DARE learned correlation

Figure 8: The ground truth (GT) and learned correlation between history behaviors of top-10 frequent categories (y-axis) at various positions (x-axis), with category 15 as the target. Our correlation scores are noticeably closer to the ground truth.

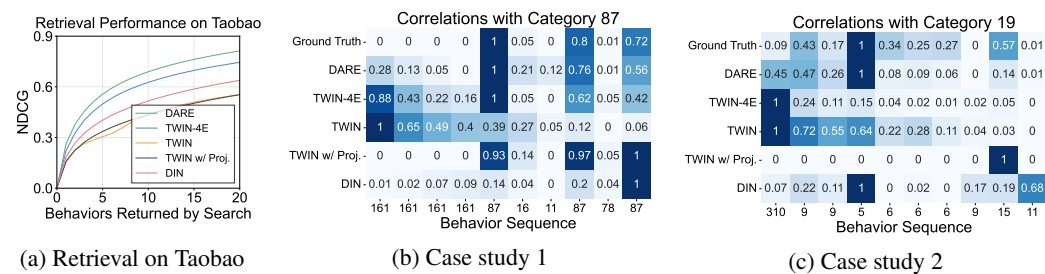

(a) Retrieval on Taobao   (b) Case study 1   (c) Case study 2

Figure 9: Retrieval in the search stage. (a) Our model can retrieve more correlated behaviors. (b-c) Two showcases where the x-axis is the categories of the recent ten behaviors.

- *DARE achieves significantly better retrieval.* As shown in Fig. 9a, the NDCG of our model is substantially higher than all baselines, with a 46.5% increase (0.8124 vs. 0.5545) compared to TWIN and a 27.3% increase (0.8124 vs. 0.6382) compared to DIN.

- *TWIN is overly sensitive to temporal information.* As discussed, TWIN tends to select recent behaviors regardless of their categories, against the ground truth, due to overestimated correlations between different categories, as shown in Fig. 9b and 9c.

- *Other methods perform unstably.* For the other methods, they filter out some important behaviors and retrieve unrelated ones in many cases, which explains their bad performance.

> **Result 1.** *DARE succeeds in capturing the semantic-temporal correlation between behaviors and the target, retaining more correlated behaviors during the search stage.*

## 4.4 REPRESENTATION DISCRIMINABILITY

We then analyze the discriminability of learned representation. On test datasets, we take the compressed representation of user history $\boldsymbol{h} = \sum_{i=1}^{K} w_i \cdot (\boldsymbol{e}_i \odot \boldsymbol{v}_t)$, which forms a vector for each test sample. Using K-means, we quantize these vectors, mapping each $\boldsymbol{h}$ to a cluster $Q(\boldsymbol{h})$. The mutual information (MI) between the discrete variable $Q(\boldsymbol{h})$ and label $Y$ (whether the target was clicked) can then reflect the representation's discriminability: $\text{Discriminability}(\boldsymbol{h}, Y) = \text{MI}(Q(\boldsymbol{h}), Y)$.

As shown in Fig. 10a, across various numbers of clusters, our DARE model outperforms the state-of-the-art TWIN model, demonstrating that decoupling improves representation discriminability. There are also other notable findings. Although DIN achieves more accurate retrieval in the search stage (as evidenced by a higher NDCG in Fig. 9a), its representation discriminability is obviously lower than TWIN, especially on Taobao dataset, which explains its lower overall performance. TWIN-4E shows comparable discriminability to our DARE model, further confirming that its poorer performance is due to inaccurate attention caused by the lack of recommendation-specific prior knowledge.

To fully demonstrate the effectiveness of $\boldsymbol{e}_i \odot \boldsymbol{v}_t$, we compare it with the classical concatenation $[\Sigma_i \boldsymbol{e}_i, \boldsymbol{v}_t]$. As shown in Fig. 10c, a huge gap (in orange) is caused by the target-aware representation,

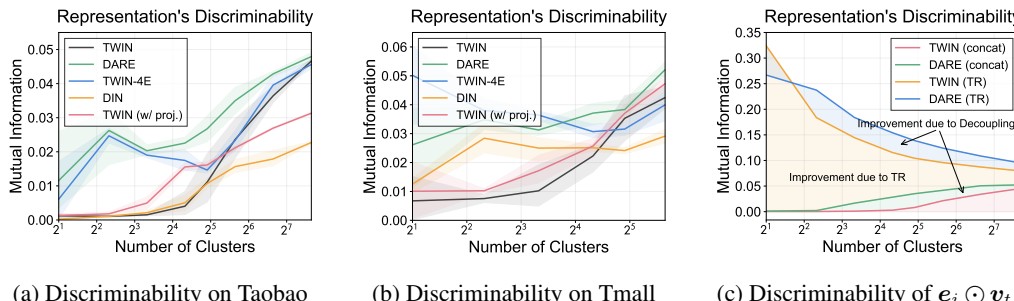

Figure 10: Representation discriminability of different models, measured by the mutual information between the quantized representations and labels.

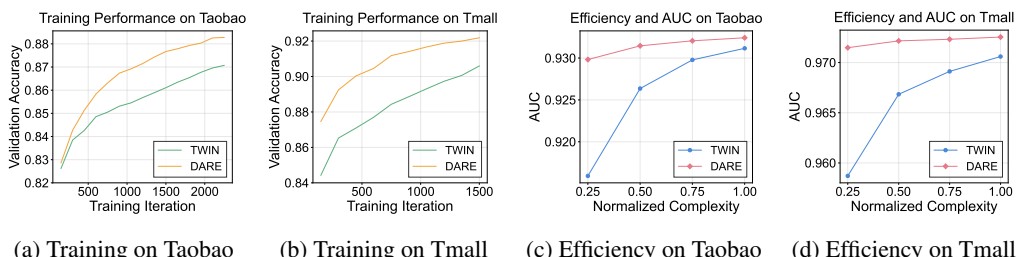

Figure 11: Efficiency during training and inference. (a-b) Our model performs obviously better with fewer training data. (c-d) Reducing the search embedding dimension, a key factor of online inference speed, has little influence on our model, while TWIN suffers an obvious performance loss.

while smaller gaps (in blue and green) result from decoupling. Notably, our DARE model also outperforms TWIN even when using concatenation.

> **Result 2.** *In the DARE model, the form of target-aware representation and embedding decoupling both improve the discriminability of representation significantly.*

### 4.5 CONVERGENCE AND EFFICIENCY

**Faster convergence during training.** In recommendation systems, faster learning speed means the model can achieve strong performance with less training data, which is especially crucial for online services. We track accuracy on the validation dataset during training, shown in Fig. 11a. Our DARE model converges significantly faster. For example, on the Tmall dataset, TWIN reaches 90% accuracy after more than 1300 iterations. In contrast, our DARE model achieves comparable performance in only about 450 iterations—one-third of the time required by TWIN.

**Efficient search during inference.** By decoupling the attention embedding space $e_i, v_t \in \mathbb{R}^{K_A}$ and representation embedding space $e_i, v_t \in \mathbb{R}^{K_R}$, we can assign different dimensions for these two spaces. Empirically, we find that the attention module performs comparably well with smaller embedding dimensions, allowing us to reduce the size of the attention space ($K_A \ll K_R$) and significantly accelerate the search stage, as its complexity is $O(K_A N)$ where $N$ is the length of the user history. Using $K_A = 128$ as a baseline ("1"), we normalize the complexity of smaller embedding dimensions. Fig. 11c shows that our model can accelerate the searching speed by 50% with quite little influence on performance and even by 75% with an acceptable performance loss, offering more flexible options for practical use. In contrast, TWIN experiences a significant AUC drop when reducing the embedding dimension.

> **Result 3.** *Embedding decoupling leads to faster model training convergence and at least 50% inference acceleration without significantly influencing the AUC by reducing the dimension of attention embeddings.*

### 4.6 ONLINE A/B TESTING AND DEPLOYMENTS

We apply our methods to Tencent's advertising platform. Since users' behaviors on ads are sparse, which makes the sequence length relatively shorter than the content recommendation scenario, we involve the user's behavior sequence from our article and the micro-video recommendation scenario. Specifically, the user's ad and content behaviors in the last two years are introduced. Before the search, the maximal length of the ads and content sequence is 4000 and 6000, respectively, with 170 and 1500 on average. After searching with DARE, the sequence length is reduced to less than 500. Regarding sequence features (side info), we choose the category ID, behavior type ID, scenario ID, and two target-aware temporal encodings, *i.e.*, position relative to the target, and time interval relative to the target (with discretization). There are about 1.0 billion training samples per day. During the 5-day online A/B test in September 2024, the proposed DARE method achieves 0.57% cost, and 1.47% GMV (Gross Merchandize Value) lift over the production baseline of TWIN. This would lead to hundreds of millions of dollars in revenue lift per year.

### 4.7 SUPPLEMENTARY EXPERIMENT RESULTS IN APPENDIX

**Retrieval number in the search stage**. DARE's advantage is more obvious with less retrieval number, proving once again that DARE selects important behaviors more accurately (Appendix D.1).

**Sequence length and short-sequence modeling**. DARE can consistently benefit from longer sequences, while it delivers marginal advantages in short-sequence modeling (Appendix D.2).

**GAUC and Logloss**. Besides AUC, we also evaluate DARE and all the baselines under GAUC and Logloss. DARE shows consistent superiority, proving the solidity of our results (Appendix E.1).

## 5 RELATED WORK

**Click-through rate prediction and long-sequence modeling.** CTR prediction is fundamental in recommendation systems, as user interest is often reflected in their clicking behaviors. Deep Interest Network (DIN) (Zhou et al., 2018) introduces target-aware attention, using an MLP to learn attentive weights of each history behavior regarding a specific target. This framework has been extended by models like DIEN (Zhou et al., 2019), DSIN (Feng et al., 2019), and BST (Chen et al., 2019) to capture user interests better. Research has proved that longer user histories lead to more accurate predictions, bringing long-sequence modeling under the spotlight. SIM (Pi et al., 2020) introduces a search stage (GSU), greatly accelerating the sequence modeling stage (ESU). Models like ETA (Chen et al., 2021) and SDIM (Cao et al., 2022) further improve this framework. Notably, TWIN (Chang et al., 2023) and TWIN-V2 (Si et al., 2024) unify the target-aware attention metrics used in both stages, significantly improving search quality. However, as pointed out in Sec. 2.2, in all these methods, attention learning is often dominated by representation learning, creating a significant gap between the learned and actual behavior correlations.

**Attention.** The attention mechanism, most well-known in Transformers (Vaswani et al., 2017), has proven highly effective and is widely used for correlation measurement. Transformers employ Q, K (attention projection), and V (representation projection) matrices to generate queries, keys, and values for each item. The scaled dot product of query and key serves as the correlation score, while the value serves as the representation. This structure is widely used in many domains, including natural language processing (Brown et al., 2020) and computer vision (Dosovitskiy et al., 2021). However, in recommendation systems, due to the interaction-collapse theory pointed out by Guo et al. (2024), the small embedding dimension would make linear projections completely lose effectiveness, as discussed in Sec. 2.3. Thus, proper adjustment is needed in this specific domain.

## 6 CONCLUSION

This paper focuses on long-sequence recommendation, starting with an analysis of gradient domination and conflict on the embeddings. We then propose a novel Decoupled Attention and Representation Embeddings (DARE) model, which fully decouples attention and representation using separate embedding tables. Both offline and online experiments demonstrate DARE's potential, with comprehensive analysis highlighting its advantages in attention accuracy, representation discriminability, and faster inference speed.

## REPRODUCIBILITY STATEMENT

To ensure reproducibility, we provide the hyperparameters and baseline implementation details in Appendix A, along with dataset details in Appendix B. We have released the full code, including dataset processing, model training, and analysis experiments, at `https://github.com/thuml/DARE`.

## ACKNOWLEDGEMENTS

This work was supported by the National Natural Science Foundation of China (62021002), the BNRist Project, the Tencent Innovation Fund, and the National Engineering Research Center for Big Data Software.

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

# A  IMPLEMENTATION DETAILS

## A.1  HYPER-PARAMETERS AND MODEL DETAILS

The hyper-parameters we use are listed as follows:

| Parameter | Value |
|---|---|
| Retrieve number | 20 |
| Epoch | 2 |
| Batch size | 2048 |
| Learning rate | 0.01 |
| Weight decay | 1e-6 |

Besides, we use the Adam optimizer. Layers of the Multi-layer Perceptron (MLP) are set as $200 \times 80 \times 2$, which is the same as Zhou et al. (2024).

These settings remain the same in all our experiments.

## A.2  BASELINE IMPLEMENTATION

Many current methods are not open-source and may focus on a certain domain. Thus, we followed their idea and implemented their method according to our task setting. Some notable details are shown as follows:

- DIN is primarily designed for short-sequence modeling, so we introduced the search stage and aligned it with long-sequence models. Specifically, while the original DIN aggregates all historical behaviors using a learned weight, our approach enables DIN to select the top-K most significant behaviors based on these weights, the same as other long-sequence modeling techniques. Note that the original DIN is impractical for long-sequence modeling, as aggregating such extensive history would result in prohibitively high time complexity.

- TWIN-V2 is specifically designed for Kuaishou, a short video-sharing app, leveraging video-specific features to optimize performance in video recommendations. However, our experiments focus on a more general scenario where only item IDs and category IDs are available. Thus, we made some necessary adjustments while retaining the core ideas of TWIN-V2. **e.g.**, TWIN-V2 would first group the videos based on the proportion a video is played, which does not have a corresponding feature in our datasets. Consequently, we grouped user history using temporal information instead. It's understandable that outside its specific domain, TWIN-V2 cannot fully realize its potential.

# B  DATA PROCESSING

**Dataset information.**   Some detailed information is shown in Table 2. We use Taobao (Zhu et al., 2018; 2019; Zhuo et al., 2020) and Tmall (Tianchi, 2018) datasets in our experiments. The proportion of active users (Users with more than 50 behaviors) in these two datasets is more than 60%, which is relatively satisfying. Note that the Taobao dataset is more complex, with more categories and more items, which is a higher challenge for model capacity.

Table 2: Some basic information of public datasets (active user: User with more than 50 behaviors).

| Dataset | #Category | #Item | #User | # Active User |
|---|---|---|---|---|
| Taobao | 9407 | 4,068,790 | 984,114 | 603,176 |
| Tmall | 1492 | 1,080,666 | 423,862 | 246,477 |

**Training-validation-test split.**   We sequentially number history behaviors from one (the most recent behavior) to T (the most ancient behavior) according to the time step. The test dataset contains

predictions of the first behaviors, while the second behaviors are used as the validation dataset. For the training dataset, we use the $(3 + 5i, 0 \leqslant i \leqslant 18)$th behavior. Models would finish predicting the $j$th behavior based on $j - 200$ to $j - 1$ behaviors (padding if history length is not long enough). Only users with behavior sequences longer than 210 will be reserved.

We make such settings to balance **the amount and quality of training data**. In our setting, *each selected user would contribute 20 pieces of data visible to our model in the training process*. Besides, we can guarantee that *each piece of test data would contain no less than 200 behaviors*, making our results more reliable. To some degree, we break the "independent identical distribution" principle because we sample more than one piece of data from one user. However, it's unavoidable since the dataset is not large enough due to the feature of the recommendation system (item number is usually several times bigger than user number), so we finally sample with interval 5, using the $((3 + 5i)th, 0 \leqslant i \leqslant 18)$ behaviors as the training dataset.

## C   THE RESEARCH PROCESS LEADING TO DARE

### C.1   OTHER DECOUPLING METHODS

Besides linear projection, we have tried many other decoupling methods before we came up with the final DARE model. Their structures are illustrated in Figure 12. Specifically:

- **Linear projection.** This is the structure referred to as TWIN (w/ proj.) in this paragraph, applying linear projection to address the conflict.

- **Item/Category/Time linear projection**. Item, category, and time features exhibit significant differences (e.g. item number is about 1,000 times larger than category number). So we tested the effectiveness of linear projections when applied to each feature individually.

- **Cate. and time linear projection**. The number of items is too large, making it too challenging for a simple linear projection to project millions of item embeddings into another space. So we designed this model and only use linear projection on category and time.

- **Larger embedding.** To enhance the capacity of linear projection while maintaining the feature dimension, we used a larger embedding dimension while keeping the output dimension of linear projection the same as other models.

- **MLP projection.** We replace the linear projection with Multilayer Perceptron (MLP), which has much stronger capacity. This experiment aims to figure out the impact of projection capacity on model performance.

- **Avoid domination.** Basing on the original TWIN model, whenever the gradient is back propagated to the embedding (we have demonstrated in Section 2.2 that gradient from representation is about five times larger than that from attention), we manually scale the gradient from attention to make its 2-norm the same as representation, which can solve the problem of domination.

### C.2   AUC RESULT

We evaluated the models on the Taobao and Tmall datasets, with the results presented in Table 3. Among the other eight models except DARE, none of them achieved consistent and significant improvements across both datasets. The Taobao dataset is notably more complex, containing nearly nine times the number of categories as Tmall. Thus, some decoupling methods showed improvements on the simpler Tmall dataset but lost effectiveness on the more complex Taobao dataset. Interestingly, while the "MLP projection" model theoretically offers greater capacity, it failed to outperform the simpler linear projection, which captured our attention. To investigate further, we examined the gradient behavior of these models.

### C.3   GRADIENT CONFLICT IN THESE MODELS

We then observed whether these models have the potential to resolve gradient conflict. **For each category**, we observed the gradients from attention and representation **at every iteration** and cal-

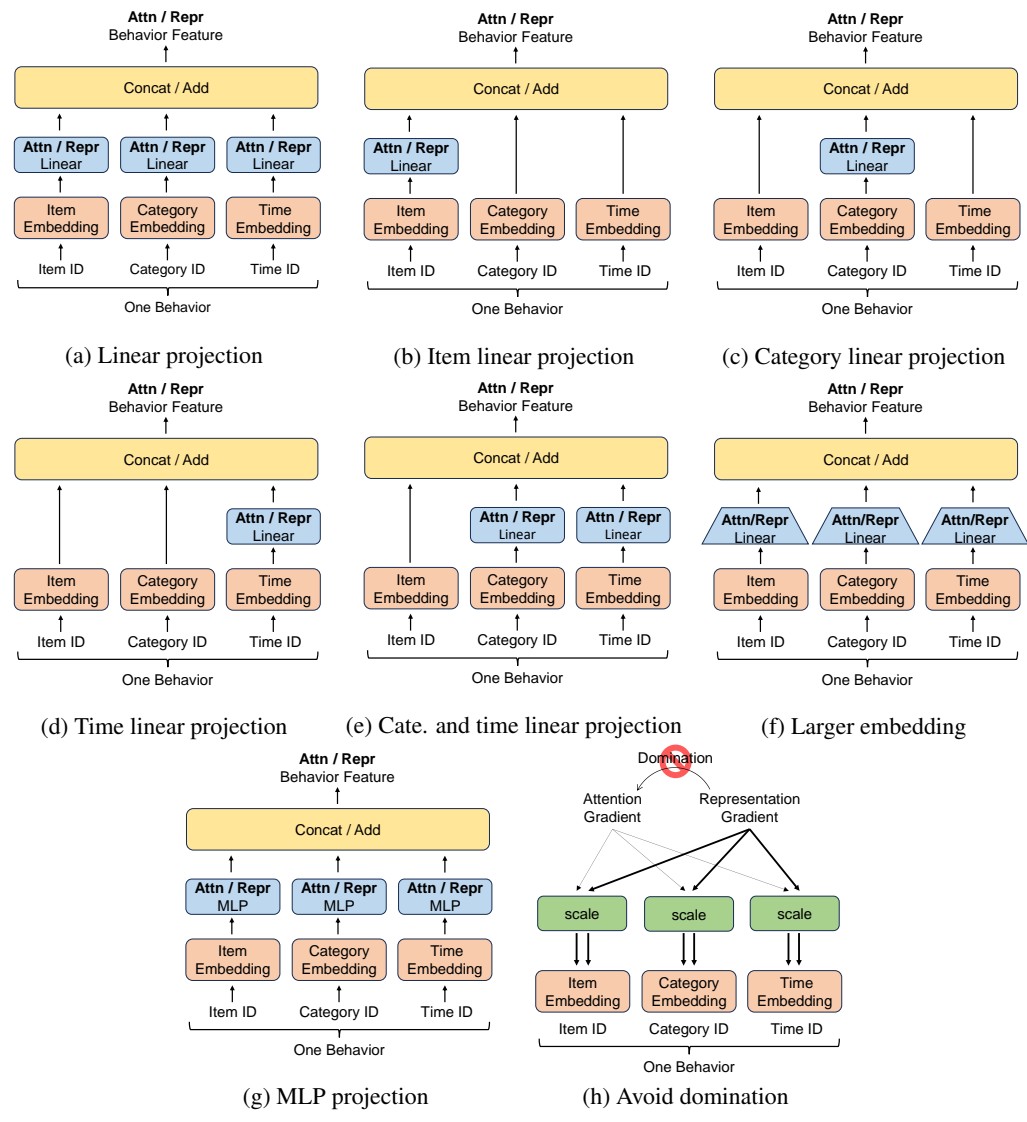

Figure 12: Eight other methods we tried before we came up with DARE.

Table 3: The performance of other models we tried reported by the means and standard deviations of AUC. Only DARE achieved a satisfying result. Each model's comparison with the original TWIN is highlighted: improvements are marked in green, while deteriorations are marked in red.

| Setup | Taobao | Tmall |
|---|---|---|
| TWIN (2021) | $0.91688 \pm 0.00211$ *(baseline)* | $0.95812 \pm 0.00073$ *(baseline)* |
| Linear projection | $0.89642 \pm 0.00351$ $(-0.02046)$ | $0.96152 \pm 0.00088$ $(+0.00988)$ |
| Item linear projection | $0.90886 \pm 0.00218$ $(-0.00802)$ | $0.96032 \pm 0.00119$ $(+0.00220)$ |
| Category linear projection | $0.91738 \pm 0.00099$ $(+0.00050)$ | $0.96658 \pm 0.00068$ $(+0.00846)$ |
| Time linear projection | $0.91354 \pm 0.00046$ $(-0.00334)$ | $0.95758 \pm 0.00116$ $(-0.00054)$ |
| Cate. and time linear projection | $0.91202 \pm 0.00340$ $(-0.00486)$ | $0.96604 \pm 0.00038$ $(+0.00792)$ |
| Larger embedding | $0.91348 \pm 0.00247$ $(-0.00340)$ | $0.96348 \pm 0.00057$ $(+0.00536)$ |
| MLP projection | $0.86960 \pm 0.04013$ $(-0.04728)$ | $0.95678 \pm 0.00066$ $(-0.00134)$ |
| Avoid domination | $0.91976 \pm 0.00052$ $(+0.00288)$ | $0.93887 \pm 0.00127$ $(-0.01925)$ |
| DARE (Ours) | $0.92568 \pm 0.00025$ $(+0.00880)$ | $0.96800 \pm 0.00024$ $(+0.00988)$ |

culated the percentage of iterations in which **the gradient for that category exhibited conflict**. Results are shown in Figure 14. As demonstrated in this figure, we can find that:

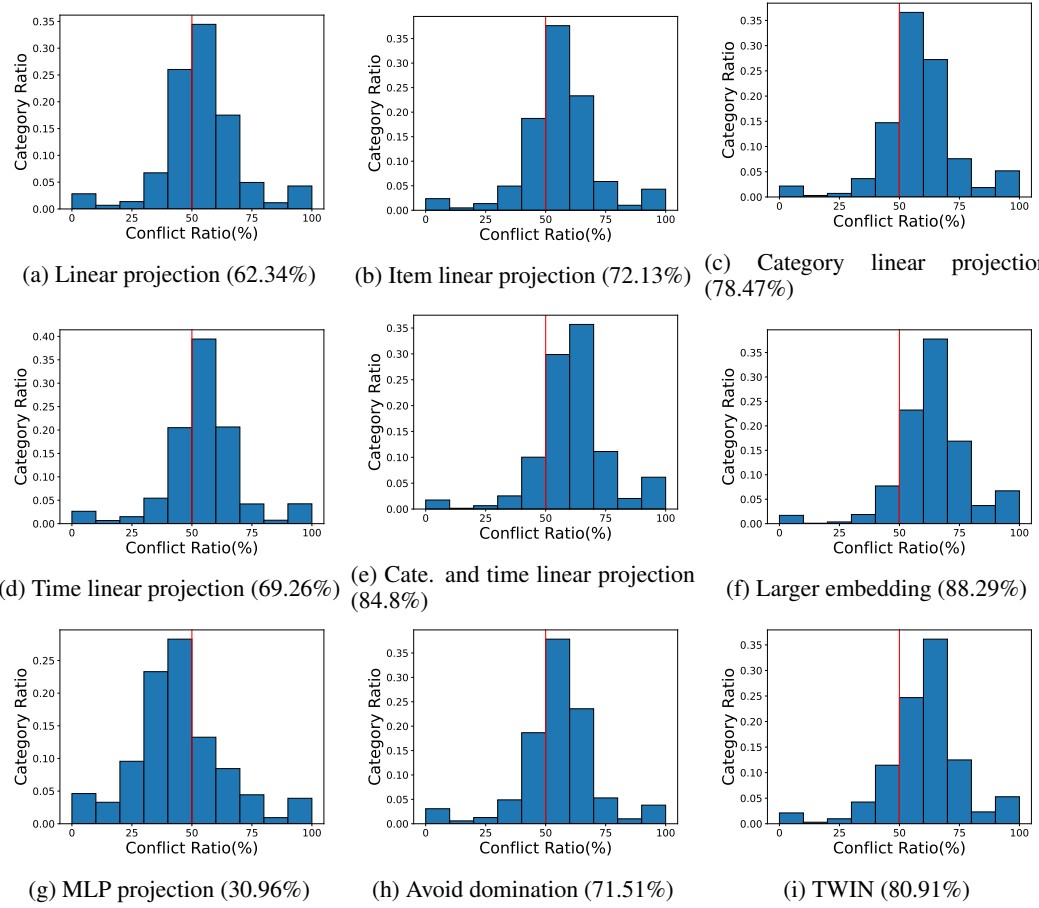

(a) Linear projection (62.34%)    (b) Item linear projection (72.13%)    (c) Category linear projection (78.47%)

(d) Time linear projection (69.26%)    (e) Cate. and time linear projection (84.8%)    (f) Larger embedding (88.29%)

(g) MLP projection (30.96%)    (h) Avoid domination (71.51%)    (i) TWIN (80.91%)

Figure 14: Analysis of gradient conflict on the original TWIN and eight other models we tried. The number after model name means the ratio of categories falling on the right side of the red line (meaning that the category reported gradient conflict in more than half iterations). Most models fail to resolve gradient conflict well.

**The conflict of TWIN.** In the original TWIN, most categories (80.91%) experienced gradient conflict in more than half of the iterations.

**The failure of these models.** Some methods (like Item linear projection) can, to some degree, solve the conflict, but that's far from enough. Some methods even worsen the conflict (like Larger embedding).

**The challenge of MLP projection.** MLP projection solves the conflict best, although still 30% categories reporting conflict in more than half of iterations, this model outperforms other projection-based decoupling methods. However, MLP projection performs poorly. To understand this discrepancy, we further analyzed its performance during training, and the results are shown in Figure 13. Though resolving conflict better than some other models, MLP projection struggles to optimize in the training process due to more parameters and higher complexity. For example, after 100 iterations, the accuracy of DARE is 82.44%, while that of MLP projection is only 74.87% (Note that even continually outputting "No" can achieve a 66.7% accuracy).

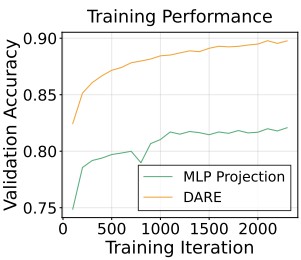

Figure 13: Comparison of MLP projection and DARE models during training.

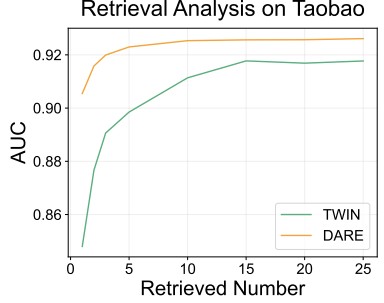 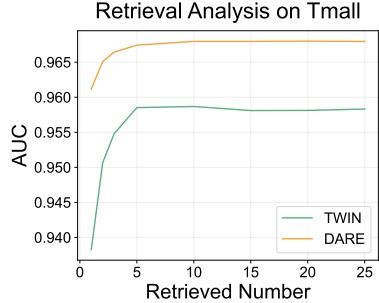

(a) Influence of retrieval number on Taobao     (b) Influence of retrieval number on Tmall

Figure 15: On both datasets, when the number of retrieved behaviors increases from 1 to 25, models first perform better, then keep the same performance. DARE outperforms TWIN at any settings, achieving even 5.7% higher AUC on Taobao when retrieving 1 behavior.

### C.4 CONCLUSION

We explored various decoupling methods, but none could fully resolve gradient conflicts, or may introduce optimization issues. All the results call for a more effective decoupling method, that is, back-propagating the gradient to different embedding tables, which can completely solve whatever problems like domination and conflict, since attention and representation will each have an exclusive embedding table now. This insight led to the development of DARE.

## D INFLUENCE OF HYPER-PARAMETERS

### D.1 EFFECTS OF RETRIEVAL NUMBER IN THE SEARCH STAGE

The number of retrieved behaviors, $K$ in this paper, is a crucial hyper-parameter in the two-stage method. We modified this parameter, and the results are presented in Figure 15. Key findings include:

- On Taobao dataset, TWIN must retrieve more than 15 behaviors to fulfill its potential, while DARE can achieve best performance when retrieving more than 10 behaviors. This indicates that DARE can retrieve those important behaviors more accurately, while TWIN must retrieve more behaviors to avoid missing important ones.

- DARE consistently outperforms TWIN across all settings, especially with fewer retrieval numbers. On Taobao dataset, when retrieving only one behavior, DARE can outperform TWIN with an AUC increase of 5.7% (even a 0.1% AUC increase is considered significant).

- In all our other experiments, the retrieve number is set to 20 to ensure all models perform at their best. Our advantage over TWIN would only be more obvious in some other settings.

### D.2 EFFECTS OF SEQUENCE LENGTH

We analyzed the impact of sequence length and identified scenarios where the DARE model exhibits a more significant advantage. Results are shown in Figure 16. Some notable findings are:

- **Reduced advantage with shorter sequences:** DARE's advantage over TWIN diminishes as the sequence length decreases. Shorter sequences make it easier to model user history, reducing the impact of inaccuracies in measuring behavior importance. Under these conditions, TWIN achieves performance comparable to DARE.

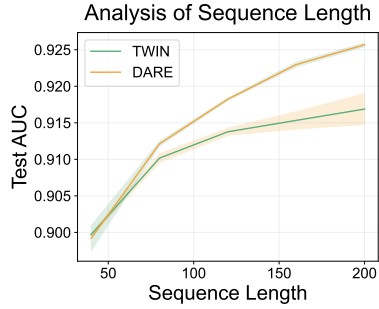
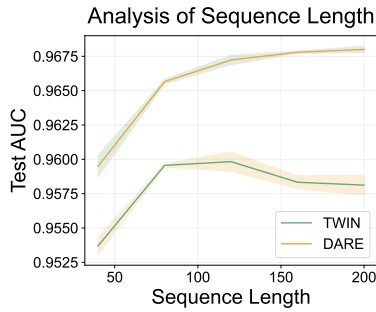

(a) Influence of sequence length on Taobao     (b) Influence of sequence length on Tmall

Figure 16: With shorter sequence length, the advantage of our DARE model over TWIN becomes smaller. DARE can perform better with longer sequence length, indicating its potential to select important behaviors in the long user history. However, on Tmall dataset, TWIN works better with sequence length 120 than 160 or 200, indicating that TWIN relies on larger embedding dimension to become effective.

- **Superior performance with longer sequences:** DARE excels with longer sequences. On the Tmall dataset **with embedding dimension=16**, however, TWIN performs worse with a sequence length of 200 compared to 120. This suggests that DARE effectively captures the importance of each behavior and leverages long user histories at any setting, while TWIN relies heavily on embedding dimension and would struggle with an abundance of historical behaviors when embedding dimension is small.

We also tried our method in the *short-sequence modeling* (removing the search stage and modeling the whole sequence). We use the Amazon dataset (He & McAuley, 2016; McAuley et al., 2015) with the same setup as the state-of-the-art TIN model (Zhou et al., 2024). However, the performance improvement is marginal (TIN: 0.86291±0.0015 AUC vs. DARE: 0.86309±0.0004 AUC). For the Amazon dataset, the average user history length is no longer than ten. Shorter sequence means fewer candidate behaviors, so it becomes easier to model behavior importance. Removing the search stage means the important behaviors will never be discarded by mistake as in long-sequence modeling, so the attention module will not cause a too severe result even if it is not capable enough. As shown by TIN Zhou et al. (2024), representation is more critical than attention in short-sequence settings, so *the dominance of representation doesn't significantly impact performance when the attention task is relatively easier.*

**Relevance to modern recommendation systems:** It is worth noting that modeling longer user histories is a growing trend in recommendation systems (Pi et al., 2020). Contemporary online systems increasingly incorporate extended user histories, making short sequence modeling less important. As this trend continues, the advantages of the DARE model will become more pronounced in today and future online systems.

### D.3 EFFECTS OF ATTENTION AND REPRESENTATION EMBEDDING DIMENSION

In general, increasing the embedding dimension improves model performance. However, in practice, limitations such as the interaction collapse theory (Guo et al., 2024) or strict time constraints make it impractical to use arbitrarily large embeddings. To address this, we analyzed various attention-representation dimension combinations, offering insights that could guide future implementations. The results are presented in Figures17. A key observation is that the representation embedding dimension has a stronger impact on model performance compared to the attention embedding dimension. This suggests that a balanced approach–using a smaller attention embedding for faster online processing and a larger representation embedding for enhanced performance–could be an optimal strategy.

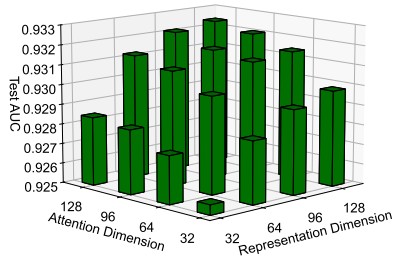 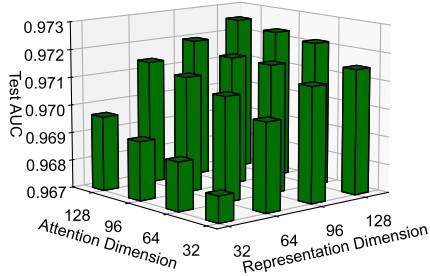

(a) AUC and embedding dim on Taobao      (b) AUC and embedding dim on Tmall

Figure 17: The influence of attention and representation embeddings on AUC.

Table 4: Overall comparison reported by the means and standard deviations of GAUC (grouped by category). The best results are highlighted in bold, while the previous best model is underlined.

| Setup | Embedding Dim. = 16 | | Embedding Dim. = 64 | | Embedding Dim. = 128 | |
|---|---|---|---|---|---|---|
| Dataset | Taobao | Tmall | Taobao | Tmall | Taobao | Tmall |
| ETA (2021) | 0.89900 (0.00372) | 0.94908 (0.00121) | 0.90980 (0.00084) | 0.95918 (0.00046) | 0.91144 (0.00050) | 0.96230 (0.00054) |
| SDIM (2022) | 0.89128 (0.00097) | 0.92408 (0.00084) | 0.89496 (0.00088) | 0.93320 (0.00067) | 0.89780 (0.00098) | 0.93062 (0.00112) |
| DIN (2018) | 0.88748 (0.00054) | 0.94998 (0.00039) | 0.89300 (0.00098) | 0.95350 (0.00039) | 0.89566 (0.00049) | 0.95628 (0.00021) |
| TWIN (2023) | 0.90364 (0.00209) | 0.94998 (0.00088) | 0.91374 (0.00065) | 0.95952 (0.00048) | 0.91880 (0.00068) | 0.96370 (0.00006) |
| TWIN (hard) | 0.89386 (0.00039) | 0.95216 (0.00037) | 0.90588 (0.00049) | 0.95698 (0.00063) | 0.90008 (0.00066) | 0.95908 (0.00021) |
| TWIN (w/ proj.) | 0.88038 (0.00405) | 0.95012 (0.00079) | 0.85372 (0.00427) | 0.94618 (0.00497) | 0.84696 (0.01012) | 0.94794 (0.00235) |
| TWIN (w/o TR) | 0.89190 (0.00081) | 0.95340 (0.00053) | 0.90098 (0.00081) | 0.95524 (0.00036) | 0.90628 (0.00095) | 0.95570 (0.00115) |
| TWIN-V2 (2024) | 0.87954 (0.00067) | 0.93772 (0.00118) | 0.88758 (0.00050) | 0.94510 (0.00038) | 0.89164 (0.00063) | 0.94894 (0.00056) |
| TWIN-4E | 0.88864 (0.01365) | 0.95278 (0.00031) | 0.88810 (0.01560) | 0.95570 (0.00051) | 0.89448 (0.01595) | 0.95148 (0.01229) |
| DARE (Ours) | **0.91240** (0.00036) | **0.96062** (0.00021) | **0.91712** (0.00052) | **0.96392** (0.00012) | **0.91966** (0.00033) | **0.96582** (0.00013) |

# E    EXTENDED EXPERIMENTAL RESULTS

## E.1    GAUC AND LOGLOSS

We also evaluated model performance using additional metrics, including GAUC (group area under the curve, grouped by category in our experiments) and Logloss (test loss). The results are presented in Tables 4 and 5. Our findings reveal that AUC and GAUC trends are consistent across all models. Logloss results largely follow the same trend, with the exception of two models: SDIM and TWIN-V2. Further analysis indicates that these two models tend to be "conservative." Let $p_+$ represent the probability of a positive outcome predicted by the model and $p_-$ represent the probability of a negative outcome. The average value of $max\{p_+, p_-\}$ is 89.55% for DARE, compared to 85.90% for SDIM and 86.78% for TWIN-V2. The prediction-confidence levels of the other seven models are similar to DARE, whereas SDIM and TWIN-V2 appear more conservative. This conservatism may help reduce their loss due to the characteristics of cross-entropy loss, but offers no tangible benefit for prediction accuracy or practical performance.

## E.2    GRADIENT CONFLICT ON TWIN

To better illustrate the universality of gradient conflict, we analyzed conflicts on a per-category basis. Specifically, each category has its own embedding (a row in the embedding table), we observed the

Table 5: Overall comparison reported by the means and standard deviations of Logloss.

| Setup | Embedding Dim. = 16 | | Embedding Dim. = 64 | | Embedding Dim. = 128 | |
|---|---|---|---|---|---|---|
| Dataset | Taobao | Tmall | Taobao | Tmall | Taobao | Tmall |
| ETA (2021) | 0.69203 (0.01410) | 0.50732 (0.00724) | 0.64648 (0.00546) | 0.44156 (0.00509) | 0.64268 (0.00692) | 0.41796 (0.00399) |
| SDIM (2022) | 0.35402 (0.00183) | 0.29586 (0.00262) | 0.34250 (0.00289) | 0.29016 (0.00270) | 0.33376 (0.00220) | 0.29238 (0.00447) |
| DIN (2018) | 0.70362 (0.00321) | 0.46560 (0.00314) | 0.68712 (0.00370) | 0.44864 (0.00287) | 0.68368 (0.00852) | 0.42956 (0.00454) |
| TWIN (2023) | 0.67436 (0.00596) | 0.49232 (0.00450) | 0.68712 (0.00370) | 0.43828 (0.00245) | 0.61176 (0.00574) | 0.40152 (0.00284) |
| TWIN (hard) | 0.66888 (0.00145) | 0.47248 (0.00303) | 0.64324 (0.00339) | 0.44020 (0.00188) | 0.63956 (0.00344) | 0.40956 (0.00166) |
| TWIN (w/ proj.) | 0.75762 (0.00854) | 0.48282 (0.00266) | 0.80758 (0.01286) | 0.50514 (0.02384) | 0.82670 (0.02353) | 0.49166 (0.01193) |
| TWIN (w/o TR) | 0.71484 (0.00368) | 0.48388 (0.00785) | 0.67618 (0.00223) | 0.47148 (0.00716) | 0.65368 (0.00542) | 0.45886 (0.00225) |
| TWIN-V2 (2024) | 0.37096 (0.00198) | 0.27066 (0.00194) | 0.35412 (0.00134) | 0.24926 (0.00187) | 0.34526 (0.00185) | 0.23646 (0.00139) |
| TWIN-4E | 0.71226 (0.03604) | 0.48144 (0.00234) | 0.70654 (0.04449) | 0.46276 (0.00164) | 0.68412 (0.04719) | 0.47432 (0.05169) |
| DARE (Ours) | 0.61922 (0.00257) | 0.41826 (0.00181) | 0.60132 (0.00341) | 0.39548 (0.00289) | 0.58960 (0.00411) | 0.38204 (0.00142) |

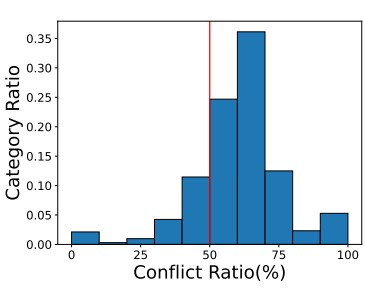

(a) Category-wise conflict in TWIN

| Conflict Ratio (%) | Average Category Frequency (%) |
|---|---|
| 0~10 | 1.63 |
| 10~20 | 6.04 |
| 20~30 | 24.15 |
| 30~40 | 45.61 |
| 40~50 | 46.11 |
| 50~60 | 64.81 |
| 60~70 | 62.61 |
| 70~80 | 38.83 |
| 80~90 | 14.52 |
| 90~100 | 2.54 |

(b) Divide categories into groups by conflict ratio. This table shows the average category frequency in each group.

Figure 18: Conflict analysis on TWIN. The category frequency is measured by the probability that a category appears in a batch.

gradient from attention and representation on this category-wise embedding. We calculated the percentage of iterations in which a conflict was reported for each category, with the results shown in Figure 18 (The same method is used in Appendix C.3). Notably, 80.91% of categories experienced conflicts in more than half of the iterations.

To explore whether conclusions like "popular categories are more likely to experience conflict" exist, we further examined the relationship between category-wise conflict ratio and category frequency. To do this, **we grouped categories based on their conflict ratios and calculated the average category popularity** (measured as the probability of a category appearing in a batch) within each group. The results are presented in Table 18b. The differences observed are largely due to statistical instability for categories that appear infrequently (for example, those categories appearing only once would have either 0% or 100% conflict ratio). However, there is no clear trend indicating that popular categories are either more or less prone to conflicts. This finding underscores the universality of gradient conflict in the TWIN model.

### E.3 LEARNED ATTENTION

Mutual Information (MI) is a measure of the amount of information that two random variables share. It quantifies the reduction in uncertainty about one variable given knowledge of another. In

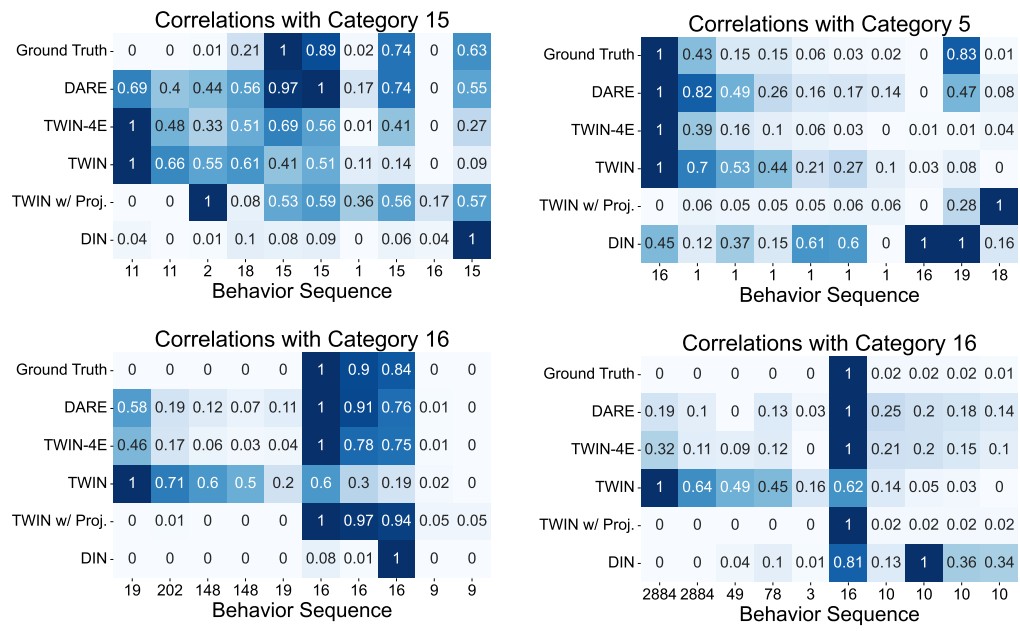

Figure 19: More case studies of the retrieval performance in search stage.

our paper, we use the standard definition of MI:

$$I(X;Y) = \Sigma p(x,y) \log \frac{p(x)p(y)}{p(x,y)}$$

where $p(x)$, $p(y)$ and $p(x,y)$ are computed based on the statistical result of the training data.

More cases of comparison between ground truth mutual information and learned attention score are shown in Figure 20. Each line contains three pictures, where the first picture is the ground truth mutual information, while the second and third line is the learned attention score of TWIN and DARE. Our DARE model is closer to the ground truth in all cases.

### E.4 Retrieval Performance during Search

More case studies of the retrieval result in the search stage are shown in Figure 19.

## F Limitation

There are also some limitations. We empirically find that linear projection only works with higher embedding dimensions, and small embedding dimensions would cause a severe "over confidence" problem. However, we still can't completely find out how this happened or what the underlying reasons are causing this strange phenomenon, which is left to future work. Besides, our AUC result in Section 4.2 indicates that target-aware representation benefits model performance in most cases, leading to an AUC increase of more than 1% on the Taobao dataset. However, on the Tmall dataset with embedding dimension = 16, TWIN (w/o TR) outperforms TWIN, which is beyond our expectations. This is possibly due to some features of the Tmall dataset (e.g. fewer items), but we could not explain this result convincingly, which is also left to future work. Finally, although two-stage methods are currently more prevalent, we also notice that there exists some one-stage methods like Yu et al. (2024). The future of these one-stage methods remains an open question, which is left for our research community.

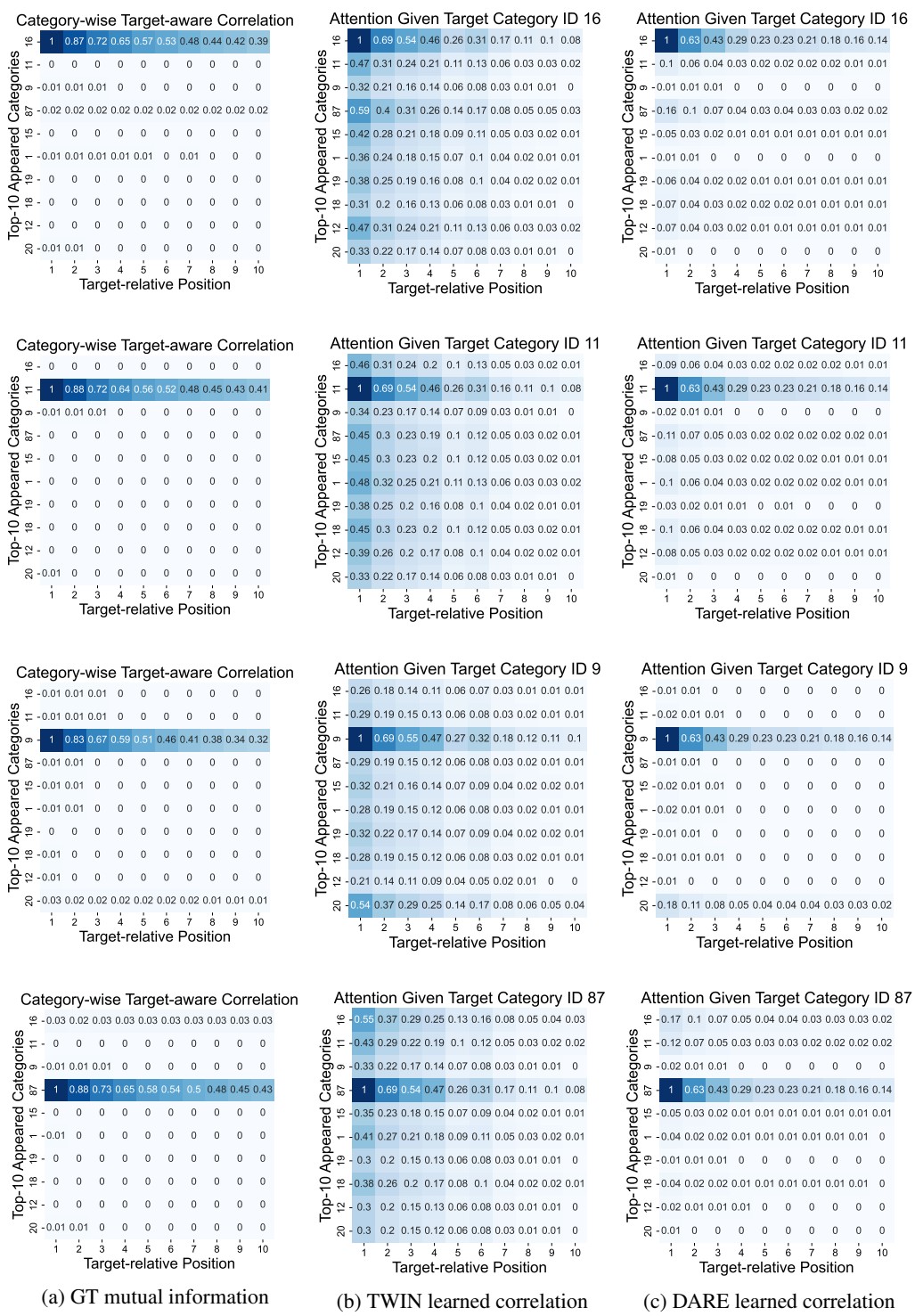

(a) GT mutual information  (b) TWIN learned correlation  (c) DARE learned correlation

Figure 20: Comparison of learned attention

