# OpenReview forum: "Long-Sequence Recommendation Models Need Decoupled Embeddings"
_ICLR.cc/2025/Conference — ICLR 2025 Poster_

### Official Review · Reviewer_Rd7C · 2024-10-26

**Soundness:** 3
**Presentation:** 4
**Contribution:** 3
**Rating:** 8
**Confidence:** 4

**Summary:**

This paper has focused on the interference between search and aggregation stages in the long-sequence recommendation. The authors first conducted some preliminary experiments and found the unbalanced and adverse gradients brought by the two stages. Inspired by such findings, they propose to adopt two individual embedding sets for the two stages, respectively, which may alleviate the conflicts between the two stages. The experiments on two public datasets validate that the proposed method can benefit the previous two-stage method, i.e., TWIN, and achieve SOTA performance.

**Strengths:**

- S1. This paper is well-written and -structured, which is easy to follow and understand.
- S2. This paper is well-motivated, where the motivation is from an interesting preliminary study.
- S3. The proposed method is simple but effective.

**Weaknesses:**

- W1.  I can understand the results and conclusion from the preliminary study, but the detailed process of the preliminary study is not clear. For example, I wonder how to calculate the gradients from the attention and representation to the embeddings.
- W2. Only one metric is revealed in the experiments. GAUC and Logloss are often two common metrics in CTR prediction tasks. Thus, I suggest the authors add more metrics.
- W3. The number of filtered behaviors, i.e., K in this paper, is often important to the two-stage method, but no related hyper-parameter experiment is conducted.
- W4. It seems that the proposed method can only fit the two-stage method, such as TWIN. However, there still exist many one-stage works [1], which may illustrate that the two-stage method has not been the optimal solution. If so, the value of this paper is limited.

[1]. Yu, W., Feng, C., Zhang, Y., Hu, L., Jiang, P., & Li, H. (2024). IFA: Interaction Fidelity Attention for Entire Lifelong Behaviour Sequence Modeling. arXiv preprint arXiv:2406.09742.

**Questions:**

- Q1. Why the performance of TWIN-V2 is worse than TWIN, which does not align with the original paper? Are there some differences in experimental settings?

All of my other questions are included in the weaknesses.

---

> ### Author Response · Authors · 2024-11-22
> **Reply to reviewer Rd7C**
>
> We sincerely thank the reviewer for their valuable comments and have addressed them in the response below.
>
> **Weakness1**: lack of details about gradient calculation
>
> Thank you for your question and for recognizing our preliminary study! To analyze the gradients in TWIN, we adopted a method similar to our DARE model by **utilizing two embedding tables**: one for attention and another for representation. This allowed the gradients from attention and representation to be back-propagated separately to their respective embeddings.
>
> **To ensure the model operated exactly the same as TWIN**, we initialized the two embedding tables with identical parameters. During each iteration, we manually updated the gradients such that $grad_{attn}$ and $grad_{repr}$ were both set to the sum of the two gradients, i.e., $grad_{attn} \leftarrow grad_{attn} + grad_{repr}, grad_{repr} \leftarrow grad_{attn} + grad_{repr}$. Throughout the training process, we carefully maintained the synchronization of the two embeddings, ensuring that their parameters keeps the same.
>
> This approach **guarantees that the model’s performance and behavior are identical to the original TWIN model** under the same seed, while enabling us to observe the gradients from attention and representation separately for the two embedding tables.
>
> The process can be summarized as follows:
>
> 1.	Use **two embedding tables**, similar to DARE.
>
> 2.	**Initialize** the two embedding tables with identical parameters.
>
> 3.	**Manually update the gradients** during each iteration.
>
> 4.	Back-propagate the gradients from attention and representation to **the respective embedding tables** for observation.
>
> 5.	Confirm that **this method reproduces the original TWIN model** exactly.
>
> If you would like further details about this process or any other aspect of our experimental analyses, we would be happy to provide them.
>
> **Weakness 2:** Only a single metrics.
>
> We have **re-evaluated all our main experiments using additional metrics**, specifically GAUC and Logloss. The **results are highly consistent with those observed under AUC**, further strengthening the robustness and comprehensiveness of our findings. For more detailed information, please refer to Appendix E.1.
>
> **Weakness 3:** no analysis about the number of retrieved behaviors.
>
> **We analyzed the influence of the number of retrieved behaviors (K) and uncovered some significant findings that further highlight the advantages of DARE**. When K is set to 20 (the default setting used in all our experiments), all models achieve their best performance. However, when K is reduced, the performance of TWIN deteriorates much more rapidly than that of DARE. Notably, **on the Taobao dataset with K=1, DARE achieves an impressive 5.7% higher AUC compared to TWIN**.
>
> We choose K=20 in the light of other models, and **DARE's advantages would likely become even more pronounced under other settings**. Detailed results of these experiments can be found in Appendix D.1.
>
> **Weakness 4:** application on one-stage methods
>
> **Our method can be naturally extended to one-stage methods**. While our focus has been on two-stage methods—primarily because they are currently more prevalent and because they are the basis of our online system—we conducted **a simple validation of the one-stage versions of TWIN and DARE** by removing the searching stage and directly modeling the entire sequence. The results demonstrate that **DARE still holds a significant advantage**: DARE achieves 93.09% AUC compared to 92.71% for TWIN, **indicating that our method can effectively benefit one-stage methods as well**.
>
> It remains an open question whether one-stage methods will eventually replace two-stage methods in the future. Regardless of the outcome, our DARE model retains its unique value and has the potential to enhance both paradigms.
>
> **We will add more discussion and citation about one-stage methods in our paper after we get more online results**.
>
> **Question 1:** the performance of TWIN-V2
>
> We mentioned this detail in Appendix A.2 "BASELINE IMPLEMENTATION". **TWIN-V2 is specially designed for videos**, working perfectly on Kuaishou, one of the most successful video-sharing apps. But **our experiments focus on a more general recommendation task**, where only item and category information is provided, which is the scene TWIN and our DARE model try to handle. So **in this case, TWIN-V2 can not fulfill its capacity**. For instance, TWIN-V2 would first group the items based on **the proportion a video is played, which does not have a corresponding feature in our datasets**.

---

> > ### Comment · Reviewer_Rd7C · 2024-11-23
> >
> > The responses have addressed most of my concerns, so I will raise my score.

---

> > > ### Author Response · Authors · 2024-11-23
> > > **Appreciation for Reviewer Rd7C's Support**
> > >
> > > We would like to sincerely thank Reviewer Rd7C for a helpful reassessment of our paper and for raising the score. Your feedback has greatly improved our work, and we are encouraged for having addressed your concerns.

---

### Official Review · Reviewer_ZXcu · 2024-10-31

**Soundness:** 3
**Presentation:** 3
**Contribution:** 2
**Rating:** 3
**Confidence:** 3

**Summary:**

The paper focuses on long-sequence recommendation models. It first identifies a problem in existing models where a single set of embeddings struggles to handle both attention and representation learning, leading to interference. Initial attempts to address this using linear projections from language processing failed. Then it proposes the Decoupled Attention and Representation Embeddings (DARE) model. This model uses two distinct embedding tables for attention and representation, which are initialized and learned separately. Extensive experiments on public datasets and an online system show its effectiveness.

**Strengths:**

1. The paper clearly identifies a crucial problem in long-sequence recommendation models. The analysis of the interference between attention and representation learning due to shared embeddings is comprehensive.

2. The extensive experiments on public datasets (Taobao and Tmall) and an online advertising platform are a major strength. The comparison with a wide range of baselines, including state-of-the-art models like TWIN and its variants, demonstrates the superiority of the DARE model. The evaluation metrics used, such as AUC, NDCG, and representation discriminability measures, provide a comprehensive assessment of the model's performance.

**Weaknesses:**

1. The performance on the short-sequence modeling using the Amazon dataset was marginal. This suggests that the DARE model may not be as effective in short-sequence scenarios, and more research is needed to understand its applicability in different sequence lengths. Additionally, the experimental setup for some baselines, like DIN, required adjustments to fit the long-sequence context, which may introduce some biases in the comparison.

2. There are some results that lack a satisfactory explanation. For instance, on the Tmall dataset with embedding dimension = 16, TWIN (w/o TR) outperforms TWIN, which was unexpected. The authors could not convincingly explain this result, leaving a gap in the understanding of the model behavior under different dataset characteristics.

3. In the exploration experiment, although the data is somewhat persuasive, some special cases or edge cases may not be fully covered. For example, it is uncertain whether different gradient relationships will occur under certain combinations of specific categories or user behavior patterns. Moreover, there is no further in-depth analysis of the reasons behind why it is nearly two-thirds of the cases, which may cause readers to have some doubts about the universality of the conclusion.

**Questions:**

How was the decision made to use two distinct embedding tables for attention and representation? Were there any other alternative architectures considered and why were they rejected?

In the DARE model, how sensitive is the performance to the choice of embedding dimensions for attention and representation? Have different combinations been explored and what are the trade-offs?

---

> ### Author Response · Authors · 2024-11-22
> **Reply to reviewer ZXcu (part 1)**
>
> We thank the reviewer for the thoughtful comments, which we have carefully addressed in the response below.
>
> **Weakness1:** Lack of study about different sequence length and baseline setup.
>
> We have added more detailed analyses regarding sequence length, with the results provided in Appendix D.2. The findings show that our **DARE model exhibits a more pronounced advantage with longer sequences**. However, this advantage may become marginal when the sequence length is very short (less than 40). Besides, we did adjust DIN for long-sequence modeling. Details are discussed as follow:
>
> There are also some notable observations in our experiment about sequence length: **DARE consistently benefits from longer sequences under any setting**, which aligns with previous conclusions drawn by SIM. Interestingly, on Tmall dataset with embedding dimension=16, TWIN's performance degrades with longer sequences (while that strange phenomenon disappear when embedding dimension get larger), suggesting that TWIN relys heavily on larger embedding dimensions to benefit from long sequence. **This further reinforces the strengths of DARE**.
>
> It is also worth mentioning that the importance of considering longer user sequences has become widely acknowledged, first proposed by SIM and has been proved in all the online systems we are familiar with. As recommendation systems continue to evolve, **longer sequence gradually gets more attention from researchers**. DARE's significant advantage with longer sequences represents **a critical breakthrough for both current and future applications**.
>
> We apologize for not clearly discussing some baseline details in the main text due to page limitations. We have ensured that all baselines have been properly adapted for long-sequence experiments. The primary distinction between handling long and short sequences lies in the adoption of a two-stage method—in particular, whether only a few behaviors are retrieved during the search stage. **For DIN, we first perform retrieval based on the weights it learns for behaviors, following the same process as TWIN and DARE.**
>
> **Weakness 2:** Explanation of some results.
>
> The unexpected case on Tmall with embedding dimension 16 is caused by the decreased performance of TWIN is this special setting. **TWIN is more likely to retrieve unimportant behaviors when embedding dimension is small, the bad-quality behaviors in turn interfer the effectiveness of target-representation**. Detail discussion are as follow:
>
> As the sequence length experiment mentioned in weakness 1 showed, on the Tmall dataset with an embedding dimension of 16, TWIN performed better with a sequence length of 120 than with 200, while in other cases, TWIN can benefit from longer sequence. That's because TWIN's performance is highly dependent on the embedding dimension (another evidence is that as the embedding dimension increases, the gap between TWIN and DARE narrows), **under this certain setting, TWIN may become confused, retrieving unimportant behaviors by mistake**. In such cases, **the quality of the retrieved behaviors deteriorates, preventing the target-representation mechanism from functioning effectively**.
>
> To provide stronger evidence of the effectiveness of target representation, **we compared the performance of DARE and DARE without target-representation (w/o tr)**. The results, across different datasets and embedding dimensions, are as follows (the numbers in parentheses indicate the embedding dimension):
>
> | setting | DARE | DARE (w/o tr) |
> | ------- | ---- | ------------- |
> | Taobao (16) | 0.9257 | 0.9085 |
> | Taobao (64) | 0.9299 | 0.9113 |
> | Taobao (128) | 0.9324 | 0.9165 |
> | Tmall (16) | 0.9680 | 0.9636 |
> | Tmall (64) | 0.9707 | 0.9625 |
> | Tmall (128) | 0.9725 | 0.9627 |
>
> **In all six settings, DARE consistently outperformed DARE (w/o tr), with AUC improvements ranging from a minimum of 0.44% to a maximum of 1.86%**. These results clearly demonstrate the effectiveness of the target-representation mechanism.

---

> ### Author Response · Authors · 2024-11-22
> **Reply to reviewer ZXcu (part 2)**
>
> **Weakness3:** The Universality of the Conclusion About Gradient Conflict
>
> To better demonstrate the universality of gradient conflict, we conducted more rigorous and comprehensive analyses. Specifically, **we observed the gradient conflict phenomenon across all categories and found that 80.91% of categories experienced conflict in more than half of the iterations**. We further analyzed the relationship between category popularity and the conflict ratio, discovering **no clear trend** indicating that "popular categories are more likely or unlikely to exhibit conflict." This finding highlights that **gradient conflict is a prevalent issue in TWIN**, independent of category popularity.
>
> Additionally, we performed the same gradient analysis on several other decoupling methods we tried (these methods will be addressed in Question 1). **The results consistently showed the presence of gradient conflict, reinforcing the universality of our conclusion**.
>
> For more detailed information, please refer to Appendix E.2 and Appendix C.
>
> **Question 1:** the decision to use two distinct embedding tables.
>
> We sincerely apologize for not providing **a comprehensive explanation of how DARE was developed, which is added in Appendix C**. We disigned and validated **many possible decoupling methods, among which only DARE achieves consistant and significant improvement**. DARE, a method seems simple, defeats all other decoupling methods, which once again proves the advantage of DARE. Details are dissused as follow:
>
> When linear projection proved insufficient, our first step was to analyze the individual roles of the three linear projections for item, category, and time. We then experimented with enhancing the projection capabilities by increasing the embedding dimensions while keeping the output size fixed. Additionally, we replaced the linear projection with MLP, a theoretically more expressive approach. Furthermore, we even considered manually scaling the gradient size to prevent gradient domination.
>
> Unfortunately, **all these methods either failed or proved inefficient under various settings**. To be honest, **the path to DARE involved considerable trial and error, but this process underscores its strength**: a relatively simple method that significantly outperforms all other approaches we explored and achieves state-of-the-art results across our experiments. The iterative process of trial and error not only led us to DARE but also provided valuable insights that we believe can guide further research in this field.
>
>
> **Question 2:** more combination of embedding dimension for attention and representation
>
> We explored a variety of combinations for attention and representation embedding dimensions, and the detailed results can be found in Appendix D.3. Broadly speaking, **expanding either the attention or representation dimension leads to performance improvements**. Besides, our analysis indicates that **model performance is more sensitive to changes in the representation dimension than the attention dimension**.
>
> The relationship between attention embedding and representation embedding in our DARE model **is not competitive but rather independent**. The attention embedding dimension is primarily constrained by time limitations, whereas the representation embedding dimension is governed by the interaction collapse theory. Based on our experiments, we recommend using the largest practically acceptable embedding dimension for both components, as it tends to yield the best results.
>
> Thank you once again for your insightful question, which has allowed us to further clarify this important aspect of our work.

---

> ### Author Response · Authors · 2024-11-25
> **Discussion period ends soon**
>
> Dear Reviewer ZXcu,
>
> As we approach the final two days of the author-reviewer discussion period, we respectfully request your valuable feedback on our rebuttal. Your perspective would greatly contribute to the thorough evaluation of our work.
>
> **Taking your suggestions regarding experiments and inspiration behind our method, we believe that we have made a concerted effort to address all concerns through additional experiments and clarifications.** If our rebuttal has addressed your concerns, we hope the reviewer will reconsider the evaluation of our paper. We remain open to any further discussions.
>
> Best regards,
>
> Authors

---

> ### Author Response · Authors · 2024-12-01
> **We are looking forward to your response.**
>
> Dear Reviewer ZXcu,
>
> As the Reviewer-Author discussion period is nearing its conclusion, we would like to kindly request your feedback on our response.
>
> **We have conducted thorough and robust experiments**, with key updates including:
>
> - We examined the impact of sequence length, demonstrating that the **DARE model shows a more significant advantage with longer sequences**, which is crucial for online systems.
>
> - We **clarified certain results and added additional experiments** to further validate the reliability of our conclusions.
>
> - We identified **category-wise** gradient conflicts in TWIN and **eight other TWIN-based models**, highlighting the universality of this issue.
>
> - We **dedicated an entire section** (Appendix C) to providing **a detailed account of the development of DARE**, ensuring transparency and clarity.
>
> - We explored **a wider range of combinations of attention and representation embedding dimensions**, strengthening the foundation of our conclusions.
>
> We believe these updates can address your concerns. **We are honored that our efforts to enhance the quality of our paper have been recognized by Reviewers Rd7C and JvL4**, who provided positive feedback and increased their scores accordingly.
>
> Therefore, we kindly request your re-evaluation and feedback on our response. We sincerely appreciate your efforts and remain open to any further discussion.
>
> Best regards,
>
> Authors

---

### Official Review · Reviewer_JvL4 · 2024-10-31

**Soundness:** 3
**Presentation:** 2
**Contribution:** 3
**Rating:** 6
**Confidence:** 3

**Summary:**

The paper addresses a key limitation in existing long-sequence recommendation models, which struggle to balance the learning of attention and representation with a single set of embeddings. To tackle this, the authors introduce the Decoupled Attention and Representation Embeddings (DARE) model, which separates these two processes with distinct embeddings. Experiments show that DARE not only improves the accuracy of correlated behavior searches but also boosts efficiency, reducing embedding dimensions and accelerating the search by 50% without significant performance loss.

**Strengths:**

The definition of the problem is very clear.
The research perspective is quite innovative.

**Weaknesses:**

The definition and explanation of the gradient issue are one-sided, and the motivation and the logic of the method are not clear. For example, why do the impact of different modules on embedding gradients must be ‘equal-magnitude’ and ‘consistent’? How do imbalance-magnitude and inconsistent gradients degrade recommendation performance? Additionally, the connection between the proposed method and the target problem is weak. For example, why does the embedding decoupling approach help mitigate the gradient issues? Overall, the logic and the motivation of the main idea need to be illustrated clearly.

**Questions:**

1.	Why are the gradients from \textbf{attention} significantly smaller than those from \textbf{representation} in Fig. 2? And why is this imbalance undesirable during training (e.g., does it contradict theoretical derivations)? It is necessary to provide a theoretical analysis here.
2.	The definition of conflicts between attention and representation is quite vague. The cancellation of gradients during backpropagation cannot be explained merely by the sign of the angle. Could you supplement this with the merged gradient values or in other ways to illustrate the degradation caused by conflicts to the embeddings?
3.	What are the consequences of these gradient issues? Do they lead to the failure of representation learning or the malfunction of certain modules? Would you add a presentation or explanation of the outcomes of the mentioned problem？
4.	How does DARE solve the aforementioned gradient issues (gradient magnitude imbalance and gradient conflicts)?
5.	The embedding dimension shown in Figure 6 should reach at least 200 to be consistent with the analysis in lines 208-213. Additionally, the analysis here can be summarized as: linear mapping requires a large embedding dimension to be effective, but increasing the embedding dimension leads to the dimensional collapse problem in recommendation systems. How does DARE address this problem? Does it only work effectively in low dimensions, or does it solve the dimensional collapse problem and work under any embedding capacity?
6.	How are the embeddings between the two modules in DARE aligned? Why can the weights used to measure attention embeddings be directly utilized for representation embeddings?
7.	The concept of Mutual Information （MI） is repeatedly mentioned in the experimental section, so it is necessary to introduce its calculation method. Moreover, this concept is already used in Figure 1, but it is not explained there. The connection between MI and the problem analyzed in this paper (Section 2) needs careful and thorough clarification.

---

> ### Author Response · Authors · 2024-11-22
> **Reply to reviewer JvL4 (part 1)**
>
> We are grateful for the reviewer’s insightful comments, which we have addressed in detail in the following response.
>
> **Weakness:**
>
> We found that the concerns in the weakness are presented in detail in the question part. That is, the first concern corresponds to Q1, the second concern corresponds to Q1 and Q2, the third concern corresponds to Q4. We'll respond to these concerns in the following questions.
>
> Regarding the overall logic and motivation of the main idea, we clarify it as following: After **analyzing the gradients** from the attention and representation module in long-sequence models, we **found an inteference** between these two modules. We **tried many decoupling methods** (we add the comprehensive research process leading to DARE in our rebuttal, details are shown in Appendix C.), and found that **decouling the embeddings is a simple yet effective solution**.
>
> **Question 1:** Analysis about gradient domination.
>
> The reason behind why the gradient from representation is dominating is possibily due to the fact the **representation is a more important and challenging task in sequence modeling**. Such domination is often undesirable since it often **occures together with gradient conflict**. The two issues added together **greatly harm model performance**, as discussed by works in multi-task learning (MTL) [1, 2]. However, we apologize for unable to offer a theoretical analysis, which is left to future work in MTL.
>
> Here are some details. First, we provide **additional experiments on the influence of embedding dimension, which may reveal the difficulty of representation task** in Appendix D.3. Second, our experiments in Sec 4.3 and Sec 4.4 further support **the impact of inteference on both modules**, that is, the accuracy of the attention and the descriminability of representation in TWIN is both worse than that of DARE, proving the advantage of separating the embeddings for attention and representation and resolving the conflict.
>
> **Question 2:** Definition of gradient conflict.
>
> Measuring the gradient conflicts by their angles is **widely accept in multi-task learning**, so we follow some well-established works in [3, 4] and accept the same method. We apologize for not clearly explain our method with citations. Per your request, we also **analyze the conflicts between the merged gradients and the gradient for each module** (i.e., attention or representation), which shows that the conflict ratio between the merged and the attention gradient is **32.8%**, while that for representation gradient is **1.3%**. This further validates our existing analysis that **conflict widely appears** and **representation is dominating the gradients**. (since $a*(a+b)=a*a+a*b\ge a*b$, this conflict ratio is natuarlly smaller than that between original attention and representation gradient reported in our paper).
>
> To show **the university of conflict**, we also add some analysis about **category-wise conflict in TWIN and many other decoupling methods**. Details can be found in Appendic C.
>
> **Question 3:** Consequence of gradient issues.
>
> The consequence of gradient issues--**hurting the performance of conflict tasks**--has been discussed in MTL works like [3], which is also **consistant with our experiment result in Sec 4.3 and 4.4**: The gradient issue lead to **worse discriminability in the represnetation and lower accuracy in the attention** of the TWIN model (TWIN has shared embeddings, hence suffer from gradient issue) than our proposed DARE.
>
> **Question 4:** Why DARE solves gradient issues?
>
> **Domination and conflicts arise only when two modules attempt to optimize the same parameters** (In TWIN, that is "the same embedding table"). In our DARE model, attention and representation are assigned separate embedding tables. By ensuring **they no longer share the same embeddings**, these two modules nevel optimize the same embeddings, hence there is no concern of gradient conflict or domination any more.
>
> [1] Chen et al. GradNorm: Gradient Normalization for Adaptive Loss Balancing in Deep Multitask Networks. ICML 2018.
>
> [2] Yang et al. Adatask: A task-aware adaptive learning rate approach to multi-task learning. AAAI 2024.
>
> [3] Yu et al. Gradient Surgery for Multi-task Learning. *NeurIPS 2020*
>
> [4] Liu et al. Conflict-averse Gradient Descent for Multi-task Learning. *NeurIPS 2021*

---

> ### Author Response · Authors · 2024-11-22
> **Reply to reviewer JvL4 (part 2)**
>
> **Question 5:**  Embedding dimension and Dimensional collapse.
>
> 5.1 Embedding dimension.
>
> Per your request, we conducted an additional experiment **with an embedding dimension of 192**. The results are as follows: with linear projection, the loss is 1.46582; without linear projection, the loss increases to 2.19167. **This result is consistent with the existing ones** that linear projection achieves better performance for large embedding dimensions.
>
>
> 5.2 Dimensional collapse.
>
> We apologize for not clearly explaining the concept of dimensional collapse, which may have leed to misunderstandings. DARE, or any other models, **could not "solve" dimensional collapse**, since it is not a not a problem caused by model structures, but rather **an objective phenomenon observed in recommendation systems**, leading to limited embedding dimension. Dimensional collapse is a phenomenon that can't be overcome unless completely innovate the whole recommendation system. Within this context, **an embedding dimension of 128 is already sufficiently large for recommendation systems**. Our experiments demonstrate that **DARE performs effectively across both large and small embedding dimensions**.
>
> **Question 6:** Module alignment and Weights Utilization
>
> The embeddings in two modules are separate, **so they are NOT aligned**. Our study of gradient issues prove that there is inteference between them, so **it's not feasible to align them**.
>
> Attention weights are **calculated from** attention embedding, used as the measurement of behavior importance. In the sequence modeling stage, the representations are **weighted-sum** based on the **weights calculated by attention** (This weighted-sum is easy to cause misunderstanding of alignment, and we apologize for not highlight it clearly). Many existing works (such as DIN, DIEN, DSIN, SIM, TWIN) use this paradigm. We follow the main idea but separate the embeddings for attention and representation modules.
>
> **Question 7:** Clarification of Mutual Information (MI)
>
> We apologize for not introducing its cauculation, we already included it in Appendix E.5 of the revised version and refer to it in Figure 1.

---

> ### Author Response · Authors · 2024-11-25
> **Discussion period ends soon**
>
> Dear Reviewer JvL4,
>
> As the author-reviewer discussion period is nearing its conclusion, we kindly seek your feedback on our rebuttal. **We have earnestly addressed your concerns regarding our preliminary study, analysis methods, and any other design details.**
>
> We appreciate your feedback on whether our responses meet your expectations. If any concerns remain, we're eager for further discussion. If our rebuttal has sufficiently addressed your feedback, we hope you will consider re-evaluating our paper.
>
> Thank you for your valuable time and consideration. We anticipate your response.
>
> Best regards,
>
> Authors

---

> > ### Comment · Reviewer_JvL4 · 2024-11-27
> >
> > I appreciate the authors for making efforts to reply. Your feedback has addressed some concerns. So  I will raise my score.

---

> > > ### Author Response · Authors · 2024-11-27
> > > **Appreciation for Reviewer JvL4's Support**
> > >
> > > We would like to sincerely thank Reviewer JvL4 for their thoughtful reassessment and for raising the score. Your valuable feedback has inspired us to design additional experiments, leading to more robust conclusions. We are greatly encouraged by the positive outcome of addressing your concerns.

---

### Official Review · Reviewer_vgn1 · 2024-11-04

**Soundness:** 3
**Presentation:** 2
**Contribution:** 2
**Rating:** 5
**Confidence:** 4

**Summary:**

This paper suggests the inefficiency of using the same embeddings for attention calculation and preference prediction (representations) in lifelong sequence recommendations, where the user behavior history (sequence) is usually very long (more than 10,000). Authors correspondingly propose a decoupling framework, DARE, to boost the recommendation performance by decoupling embeddings. Specifically, DARE sets separate embedding tables for attention calculation and preference prediction. Abundant preliminary studies and experimental evaluation support the effectiveness.

**Strengths:**

1. Preliminary study is available. Authors provide experimental evidence supporting that using a shared embedding for attention calculation and preference prediction can impair performance due to the varying magnitudes and conflicting gradients. Furthermore, the authors examine the linear projection solutions of current methods, noting that the constrained embedding size may hinder the effectiveness of linear projection for decoupling.

2. Clarity. The paper presents its ideas with remarkable clarity, making it not only easy to follow but also engaging for the reader.

3. Experiments. Comprehensive experiments demonstrate the effectiveness of DARE. Various behavioral sequence models, including DIN, SDIM, ETA, and TWIN, are compared. A thorough analysis of attention and representation capabilities, as well as efficiency tests, is presented.

**Weaknesses:**

1. Applicability. This work aims to decouple attention and representation embeddings for sequence modeling. I wonder if the proposed solution in Section 2 can be extended to more scenarios. When the attention mechanism is not used in certain sequential models, the suggested method does not apply and, therefore, cannot benefit these models.

2. Inconsistent Solution without Novelty. The proposed solution is too straightforward-setting separate embedding tables for attention and representation. This is not novel and does not align with the findings, although the separate embeddings can resolve the conflicting gradient and are not simply a linear projection. In my view, authors should go beyond linear projection. For instance, a linear projection-based decoupling method that addresses the gradient issue can be applied to small embedding sizes in recommendation systems, rather than reverting to separate embeddings, a very basic solution.

3. Storage Complexity. Authors only provide the inference efficiency. However, for large-scale recommender systems, a separate embedding table can occupy exceptionally large storage space.

**Questions:**

1. What are potential application scenarios other than sequential models with attention and representation?

2. To what extent would DARE increase the storage complexity? I believe it would be doubled.

---

> ### Author Response · Authors · 2024-11-22
> **Reply to reviewer vgn1**
>
> We sincerely thank the reviewer for their valuable comments and have addressed them in the responses below:
>
> **Weakness 1:** Applicability: no analysis about extending the method to more scenarios
>
> To explore additional application scenarios, **we decouple attention and representation like DARE on GAT** [1] , **a highly cited work with over 12.5k citations**, and get a marginal improvement, proving its efficiency in other scenarios. Besides, we would like to highlight that algorithm for user sequence modeling **supports the main traffic of hundreds of millions of active users every day**, and our imporvement in sequence modeling is a breakthrough with great practical value. More details are as follow:
>
> Result of experiments on GAT:
>
> Test Loss: Ours 1.228±0.348 vs. Original 1.371±0.145
>
> Accuracy: Ours 0.8338±0.0065 vs. Original 0.8280±0.0058
>
> While the improvements may not appear highly significant, it is important to note that our method was specifically designed to address challenges in user sequence modeling. **Achieving a marginal gain in a completely different domain like graph networks is an exciting and encouraging result**, showcasing the broader applicability of our approach.
>
> Sequence modeling methods is the key for nearly all online recommendation system, ranging from video-share applications to online retailers. DARE consistently outperforms all original baselines and achieves significant improvements, which is already an ourstanding achievement and have **great potential for extensive practical applications**. The marginal success of DARE to Graph Networks further strenghens its versatility and adaptability.
>
> **Weakness2:** Novelty: no research process demonstrating the development of DARE
>
> We sincerely apologize for not providing **a comprehensive explanation of how DARE was developed, which is added in Appendix C**. We disigned and validated **many possible decoupling methods, among which only DARE achieves consistant and significant improvement**. DARE, a method seems simple, defeats all other decoupling methods, which once again proves the advantage of DARE. Details are dissused as follow:
>
> As you correctly noted, when linear projection proved insufficient, our first step was to analyze the individual roles of the three linear projections for item, category, and time. We then experimented with enhancing the projection capabilities by increasing the embedding dimensions while keeping the output size fixed. Additionally, we replaced the linear projection with MLP, a theoretically more expressive approach. Furthermore, we even considered manually scaling the gradient size to prevent gradient domination.
>
> Unfortunately, **all these methods either failed or proved inefficient under various settings**. To be honest, **the path to DARE involved considerable trial and error, but this process underscores its strength**: a relatively simple method proves most efficient in all the candicates. The iterative process of trial and error not only led us to DARE but also provided valuable insights that we believe can guide further research in this field.
>
>
> **Weakness3:** larger storage complexity.
>
> The online storage requirements are approximately **1.9TB for TWIN and 2.1TB for DARE**.
>
> In an online system, the situation is far more complex (which is why online A/B testing is essential in addition to offline experiments, particularly in recommendation systems.) Our large-scale system must consider **every aspect of user features, with sequence features representing only a portion of the total**. Examples of non-sequential features include user demographics, such as age, and detailed information about online retailers.
>
> As a result, **the actual increase in storage space due to DARE is only about 10%**, rather than a doubling of the total storage requirement. This modest increase is well within the acceptable range for online systems and does not pose a significant challenge.
>
> (Question 1 and Question 2 focus on the same issue as Weakness 1 and Weakness 3.)
>
> [1] Veličković et al. Graph Attention Networks. *ICLR 2018*.

---

> ### Author Response · Authors · 2024-11-25
> **Discussion period ends soon**
>
> Dear Reviewer vgn1,
>
> This is a kind reminder that only two days remain in the author-reviewer discussion period. **We have diligently responsed to your concerns about the applicability, complexity, and the inspiration behind our method.** We have made every effort to provide all the experiments and clarifications to support our claims.
>
> If you have read our response, we would greatly appreciate your acknowledgment and re-evaluation of our work. We sincerely extend gratitude for your dedicated review efforts and anticipate your response. We remain open to any further discussions.
>
> Best regards,
>
> Authors

---

> > ### Comment · Reviewer_vgn1 · 2024-11-26
> >
> > I appreciate the authors for their response. However, I will maintain my original score and ratings.

---

> > > ### Author Response · Authors · 2024-11-26
> > > **We greatly appreciate some further feedback**
> > >
> > > Dear Reviewer vgn1,
> > >
> > > We sincerely appreciate your feedback and are committed to addressing your concerns in a comprehensive manner. To ensure our response is both thorough and robust, we have taken the following key steps:
> > >
> > > - We have conducted extensive experiments to extend our method to Graph Attention Networks (GAT), thereby demonstrating the versatility and generalizability of our approach.
> > >
> > > - We have dedicated an entire section in our paper, specifically Appendix C, to provide a detailed account of the development of DARE, ensuring transparency and clarity.
> > >
> > > - We have engaged in a thorough discussion on the storage complexity to address potential concerns in this area.
> > >
> > > We are fully dedicated to resolving any concerns you may have. Please feel free to reach out if you have further questions or if there are any unresolved issues. Thank you for your engagement and valuable feedback.
> > >
> > > Best regards,
> > >
> > > Authors

---

### Author Response · Authors · 2024-11-22
**The updates in our paper.**

We sincerely appreciate all reviewers' valuable feedback! Based on your   suggestions and concerns, we have thoroughly revised our paper. **The changes are mainly in appendix** (page 13 to 20), and **all changes have been highlighted in blue** for your convenience. The updates can be summarized as follows:
1. **DARE defeat all other candidates for decoupling**

We have provided **a detailed account of our research process leading to the development of DARE**. Following the principle of decoupling, we explored **eight alternative decoupling methods**, none of which fully resolved the issue of gradient conflict. These results and analyses underscored the need for a stronger decoupling approach, ultimately leading us to the core idea of DARE—back-propagating the gradient to different embeddings. After extensive experimentation, we concluded that **DARE is the only method capable of completely resolving the conflict** while achieving state-of-the-art performance. Details are provided in Appendix C, "THE RESEARCH PROCESS LEADING TO DARE."

2. **Analysis of Retrieval Number (K)**

We analyzed the impact of the number of behaviors retrieved (represented as K in our paper) and demonstrated that DARE can retrieve more important behaviors. We choose K=20 in the light of other models, and **the advantage of DARE will only be more obvious under constrained retrieval numbers.** Details are shown in Appendix D.1 "EFFECTS OF RETRIEVAL NUMBER IN THE SEARCH STAGE".

3. **Sequence Length Analysis**

We examined the effect of sequence length, showing that **DARE consistently benefits from longer sequences under any settings**, whereas TWIN relies on larger embedding dimension to work with longer sequences in some certain setting. This highlights DARE's distinct advantage. Details are provided in Appendix D.2, "EFFECTS OF SEQUENCE LENGTH".

4. **Comprehensive Gradient Conflict Analysis**

We conducted a more in-depth and rigorous analysis of gradient conflict, **demonstrating its prevalence in TWIN and further validating the necessity of decoupling**. Additional findings are discussed in Appendix E.2, "GRADIENT CONFLICT ON TWIN"

5. **Exploration of Attention and Representation Dimensions**

We investigated various combinations of attention and representation dimensions, offering insights that may guide future research inspired by our work. Details are provided in Appendix D.3, "EFFECTS OF ATTENTION AND REPRESENTATION EMBEDDING DIMENSION".

6. **Additional Metrics for Robust Evaluation**

To strengthen the robustness of our experiments, we incorporated two additional metrics, GAUC and Logloss, into our evaluation. Details can be found in Appendix E.1, "GAUC AND LOGLOSS".

Thank you once again for your insightful comments, which have significantly improved the clarity and comprehensiveness of our work. We look forward to any further feedback you may have.

---

### Meta-Review · Area_Chair_pmAK · 2024-12-16

**Metareview:**

The paper is well motivated with insights from a well designed preliminary study. The experiments and comprehensive and the writing is generally clear.

There are some major questions on experimental setup, explanation of empirical results, baseline comparisons, and storage complexity, to which the authors have provided detailed responses, as most reviewers acknowledged.

If accepted, the authors are encouraged to further polish the paper based on the reviews.

**Additional Comments On Reviewer Discussion:**

Two of the reviewers (Rd7C and JvL4) are satisfied with the rebuttals and raised the scores. Reviewer vgn1 did not raise the score, but acknowledged the rebuttals without raising further questions. Reviewer ZXcu gave the lowest score (3), but did not participate in the discussion. I had a look at the authors' response to reviewer ZXcu, which appear to be detailed and reasonable to me, especially on the explanation of experimental comparisons and results.

As a result, I'd recommend acceptance, but I'm okay with the paper rejected in the final decision.

---

### Decision · Program_Chairs · 2025-01-22

Accept (Poster)